# GraphGPT: Generative Pre-trained Graph Eulerian Transformer

Qifang Zhao [1]  Weidong Ren [1]  Tianyu Li [1]  Hong Liu [1]  Xingsheng He [1]  Xiaoxiao Xu [1]

## Abstract

We introduce *GraphGPT*, a novel self-supervised *generative pre-trained* model for graph learning based on the *Graph Eulerian Transformer* (**GET**). First, we propose GET, which combines a standard transformer encoder or decoder architecture with an innovative graph-to-sequence transformation method. This method converts graphs or sampled subgraphs into sequences of tokens representing nodes, edges, and attributes in a reversible manner using Eulerian paths. We pre-train GET using either of the two self-supervised tasks: next-token prediction (NTP) and scheduled masked-token prediction (SMTP). The pre-trained model is then fine-tuned for downstream tasks such as graph-, edge-, and node-level prediction. Despite its simplicity, GraphGPT achieves performance comparable to or surpassing state-of-the-art methods on multiple large-scale Open Graph Benchmark (OGB) datasets. It demonstrates exceptional results on the molecular property prediction dataset PCQM4Mv2 and the protein-protein interaction dataset ogbl-ppa. Notably, generative pre-training enables scaling GraphGPT to 2 billion parameters while maintaining performance gains — a breakthrough that overcomes the scalability limitations of traditional Graph Neural Networks (GNNs) and prior graph transformers (GTs). To advance research in graph foundation models and facilitate scientific discovery in chemistry, materials science, and related fields, we have released the source code[1] and model checkpoints[2].

[1]Alibaba Inc., Hangzhou, China. Correspondence to: Qifang Zhao <james.zqf@alibaba-inc.com>, Xiaoxiao Xu <xiaoxiao.xuxx@alibaba-inc.com>.

*Proceedings of the $42^{nd}$ International Conference on Machine Learning*, Vancouver, Canada. PMLR 267, 2025. Copyright 2025 by the author(s).

[1]https://github.com/alibaba/graph-gpt

[2]https://www.modelscope.cn/organization/Alibaba-DT

## 1. Introduction

The deep learning revolution sparked by AlexNet (Krizhevsky et al., 2012) has driven remarkable progress in computer vision (CV) and natural language processing (NLP). The graph learning community similarly shifted from traditional machine learning to deep learning with the rise of graph neural networks (GNNs) (Kipf & Welling, 2017; Hamilton et al., 2017; Zhang & Chen, 2018; Wu et al., 2020).

Today, transformers dominate CV (Dosovitskiy et al., 2021; Liu et al., 2021) and NLP (Devlin et al., 2019; Radford et al., 2018), scaling to billions of parameters (Liu et al., 2021; Brown et al., 2020) and achieving superhuman performance on benchmarks like ImageNet (Deng et al., 2009) and GLUE (Wang et al., 2019). These advances underpin transformative applications such as ChatGPT (Open-AI, 2023) and Midjourney (Midjourney, 2023).

Despite progress, GNNs remain constrained by over-smoothing (Rusch et al., 2023) and over-squashing (Alon & Yahav, 2021), limiting their scalability and capacity to leverage large-scale graph data. Recent efforts to adapt transformers to graphs (Ying et al., 2021; Kim et al., 2022; Luo et al., 2023; Müller et al., 2024) show promise but face critical challenges: 1). *Structural Bias*: Most graph transformers (GTs) rely on handcrafted features or GNN modules to encode graph topology, compromising generalization. 2). *Task Limitations*: GTs excel at graph-level tasks but struggle with edge- and node-level objectives (Müller et al., 2024). 3). *Pre-Training Gap*: Unlike NLP's success with self-supervised pre-training (Radford et al., 2018; Devlin et al., 2019), GTs lack effective frameworks for generative pre-training (Min et al., 2022; Müller et al., 2024).

In this work, we propose *GraphGPT*, a novel model for graph learning comprising three key innovations. 1). *GET Backbone*: A transformer-based architecture that operates on graph-equivalent token sequences via Eulerian paths, 2). *Self-Supervised Pre-Training*: Utilizing NTP and SMTP tasks (Radford et al., 2018; Chang et al., 2022), and 3). *Task-Agnostic Fine-Tuning*: Adapting the pre-trained model to supervised graph-, edge-, and node-level tasks.

Our contributions are summarized as follows:

- **Graph Eulerian Transformer (GET)**: We introduce GET, a novel architecture that leverages Eulerian or semi-Eulerian paths[3] to losslessly and reversibly convert graphs into token sequences. By integrating subgraph sampling and node identity encoding, GET efficiently processes graphs of arbitrary sizes. A standard transformer encoder or decoder is then applied to these sequences, eliminating the need for specialized architectural modifications.

- **Generative Pre-Training Framework**: GraphGPT is pre-trained using NTP or SMTP tasks, offering three advantages: $a$) Captures structural and semantic graph patterns without handcrafted features or domain-specific architectures, $b$) Scales to over 2 billion parameters with sustained performance gains, and $c$) Enables effective graph generation through its sequence-based formulation.

- **Unified Task Formatting**: We design a novel method to reformat graph-, edge-, and node-level tasks into sequences compatible with transformers. This approach allows downstream tasks to fully exploit pre-trained representations while unifying pretext and target task frameworks.

- **State-of-the-Art (SOTA) Performance**: Extensive experiments on OGB datasets demonstrate GraphGPT's superiority: it achieves SOTA results in graph- and edge-level tasks (e.g., molecular property prediction on PCQM4Mv2 and protein-protein interaction on ogbl-ppa), while delivering competitive performance in node-level tasks.

## 2. Approach

### 2.1. Overview

GraphGPT employs a three-stage framework: 1). *Graph-to-Sequence Transformation* of GET: The input graph is converted into a sequence of tokens via (semi-)Eulerian paths, ensuring a lossless, reversible mapping between the graph and its sequential representation. This transformation preserves node, edge, and attribute information while enabling compatibility with transformer architectures. 2). *Self-Supervised Pre-Training*: A standard transformer backbone (e.g., Llama; Touvron et al. (2023)) processes these sequences using the tasks NTP or SMTP (Radford et al., 2018; Chang et al., 2022). These tasks enable the model to learn structural and semantic graph patterns without task-specific supervision. 3). *Task-Specific Fine-Tuning*: The pre-trained model is adapted to downstream tasks—including graph

classification/regression, link prediction, and node classification—by reformatting task objectives into sequence-based inputs. This unified approach maximizes the transfer of pre-trained knowledge.

### 2.2. Graph to Sequence of Tokens

To convert graphs into token sequences, we employ distinct strategies based on graph size:

- *Small graphs* (e.g., molecular graphs) are directly serialized using the method in §2.2.1.

- *Large graphs* (with up to billions of nodes/edges) are first decomposed into subgraphs via the sampling process described in §2.2.2. Node identity preservation (§2.2.3) ensures structural consistency during this decomposition. These subgraphs are then serialized using §2.2.1's guidelines.

#### 2.2.1. SERIALIZING GRAPHS WITH (SEMI-)EULERIAN PATHS

We propose a *lossless, reversible graph serialization* method based on traversing all edges and nodes via *(semi-)Eulerian paths*. This approach guarantees:

- *Complete representation* of nodes and edges in the sequence.

- *Bijective mapping* between the graph and its serialized form[4].

**Algorithmic Foundation**: The problem aligns with the Chinese Postman Problem (Mei-Ko, 1962; Edmonds & Johnson, 1973), which seeks the shortest path traversing all edges. For graphs lacking (semi-)Eulerian properties, we apply Eulerization (Edmonds & Johnson, 1973; Daubechies & Hughes, 2009), duplicating minimal edges to create an Eulerian multigraph.

**Implementation**:

1. *Connectivity Check*:

- Check if the graph is connected. If not, link disconnected components by adding synthetic edges between randomly selected nodes. For example, given a graph with disconnected components A, B, and C, connect A and B via a random node pair, then B and C similarly.

- Label these edges with a dedicated [EDGE_JUMP] token and default attribute tokens.

---

[3]For a quick recap, a connected graph with every node of even degree is Eulerian, and with exactly two nodes of odd degree is semi-Eulerian. In this paper, '(semi-)Eulerian' refers to both Eulerian and semi-Eulerian unless specified otherwise.

[4]Given a fixed starting node and predetermined node indices, the Eulerian path generated by NetworkX (Hagberg et al., 2008) is guaranteed to be unique.

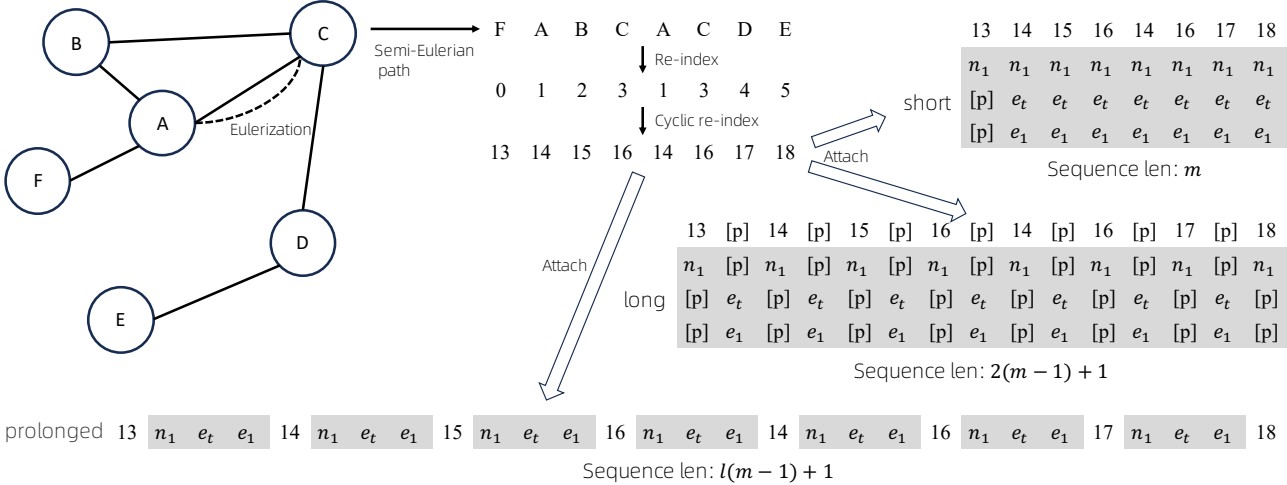

*Figure 1.* Overview of Graph-to-Sequence Tokenization. (Left) The process of converting a (sub)graph into a token sequence via a (semi-)Eulerian path. Dashed lines indicate duplicated edges added during Eulerization to enable full edge traversal. (Right) Three methods (short, long, prolonged) for integrating node/edge attributes into the Eulerian sequence. For simplicity, we assume one node attribute ($n_1$) and one edge attribute ($e_1$) per edge. Special tokens include padding token [p] and edge type $e_t$ (e.g., incoming/outgoing direction). Sequence parameters: $m$ is length of the Eulerian sequence, and $l = 2 + $ #edge-attrs + #node-attrs (here, $l = 4$).

- This ensures the graph becomes connected, enabling Eulerian path generation.

2. *Path Identification*:

- Check Eulerian properties using established criteria (West et al., 2001).

- If non-Eulerian, perform Eulerization to enable path traversal.

- Randomly sample one valid path from possible candidates, introducing stochasticity as a data augmentation strategy akin to computer vision techniques (Perez & Wang, 2017). This stochasticity forces the transformer to learn invariance across different paths of the same graph and empirically reduces overfitting.

3. *Node Re-indexing*:

- Assign indices $0, 1, \cdots, n - 1$ based on nodes' first appearance in the path (e.g., Fig. 1: node $F \to 0$, $A \to 1$).

- Introduce *cyclic re-indexing*: $i' = (i + r)\%N$, where $r$ is a random integer and $N$ (hyperparameter) exceeds the maximum node count. Without cyclic re-indexing, Eulerian paths would always start with low-index tokens (e.g., 0, 1, 2), leading to skewed token frequency distributions. Cyclic re-indexing randomizes starting indices (e.g., selecting from $\{0, 1, \cdots, 255\}$ for

$N = 256$), ensuring uniform training across all index tokens.

- Cyclic re-indexing is critical for datasets like Triangles (§3.2.1), where test graphs have significantly more nodes than training graphs (e.g., test graphs up to 100 nodes vs. training graphs $\leq$ 25 nodes). Without re-indexing, higher-index tokens (e.g., $25 \sim 255$) remained undertrained, degrading performance.

4. *Attribute Attaching*:

- Discrete attributes: Direct tokenization.

- Continuous attributes: Digit-wise tokenization (e.g., $3.14 \to [3, ., 1, 4]$).

- Edge directionality: Distinct tokens for incoming/outgoing edges (e.g., $[\to]$, $[\leftarrow]$).

- Three attribute integration strategies (Fig. 1): *short*, *long*, and *prolonged* formats.

**Theoretical Guarantee:**

The serialization is *lossless and reversible* (up to isomorphism) per the Eulerian Path Theorem: reconstructing edges from adjacent tokens recovers the original graph structure (Grohe & Schweitzer, 2020). For example, in Fig. 1's sequence, connecting consecutive tokens ($13 \to 14 \to 15 \to$ ...) reconstructs all edges, yielding a graph isomorphic to the input.

### 2.2.2. SUBGRAPH SAMPLING

Directly serializing large graphs into sequences via the method in §2.2.1 produces excessively long sequences that exceed transformer context limits. While truncating such sequences is possible, this approach faces two critical issues:

- *Computational Overhead*: Eulerization and path identification for a large graph are computationally expensive.

- *Inconsistent Training*: Sequence fragmentation introduces mismatches between pre-training and fine-tuning data formats.

To address these challenges, we adopt subgraph sampling—a scalable strategy that decomposes a large graph into smaller, manageable subgraphs before serialization.

**Implementation**:

- We use the ShaDowKHop sampler (Zeng et al., 2021) to extract localized subgraphs centered on randomly selected nodes or edges.

- Sampler parameters (e.g., hop depth, neighbor count) are preconfigured to ensure generated sequences fit within the transformer's context window. These parameters are dataset- and hardware-dependent (see App. A.1 for configuration details).

### 2.2.3. NODE IDENTITY ENCODING

Preserving global node identities during subgraph sampling is essential to avoid information loss. While unique token-based encoding (via learnable embeddings) is theoretically viable, it becomes impractical for graphs with billions of nodes due to:

- *Vocabulary Explosion*: A 10-billion-node graph would require a vocabulary of size $10^{10}$.

- *Memory Constraints*: Corresponding embedding matrices become prohibitively large.

*Solution: Multi-Token Node Encoding.* We propose encoding each node as a combination of $k$ tokens, reducing vocabulary size exponentially. For example: A $10^{10}$-node graph can be uniquely represented with two tokens from a $10^5$-size vocabulary ($10^5 \times 10^5 = 10^{10}$). Graph partitioning via METIS (Karypis & Kumar, 1997) enables this by dividing the graph into $10^5$ clusters, each containing $\sim 10^5$ nodes.

*Trade-offs*: Increasing $k$ (e.g., $k = 5$) allows smaller vocabularies ($100^5 = 10^{10}$) but lengthens sequences. This mirrors variable-length encodings like utf-8 (Allen et al., 2012, Chapter 2), where characters are represented by 1–4 bytes (vocabulary size $= 256$).

Our ablation studies (§3.5.3) demonstrate this method's effectiveness in preserving node identity.

## 2.3. Modeling with the Transformer Decoder/Encoder

We demonstrate how the transformer architecture processes graph token sequences under a unified pre-training and fine-tuning paradigm for diverse graph tasks.

### 2.3.1. PRE-TRAINING WITH THE NTP OR SMTP TASKS

Self-supervised pre-training has proven critical for success in NLP (Devlin et al., 2019; Radford et al., 2019) and CV (He et al., 2022; Chang et al., 2022). We adopt two foundational generative tasks: NTP, which enables SOTA performance in NLP (Brown et al., 2020), and SMTP, which extends masked prediction with scheduled masking rates.

**Implementation**:

- *Masking (SMTP only)*: For node-level masking, all occurrences of a masked node in the Eulerian sequence are hidden to prevent leakage (*e.g.*, two occurrences of node $A$ in Fig. 1 are masked concurrently).

- *Mask Scheduling (SMTP only)*: Following Chang et al. (2022), we adopt the same linear scheduling function, which empirically balances training stability and performance.

- *Multi-Token Prediction (NTP and SMTP)*: For sequences encoded in *short* or *long* formats (Fig. 1), we predict all non-padding tokens per column simultaneously, similar to Gloeckle et al. (2024).

### 2.3.2. FINE-TUNING ON DOWNSTREAM GRAPH TASKS

We adapt the pre-trained transformer to supervised tasks by reformatting the sequence-based inputs, ensuring alignment with pre-training:

1. *Graph-Level* Tasks (e.g., classification/regression): Append a special [GSUM] token to the sequence.

2. *Edge-Level* Tasks (e.g., link prediction): Append tokens of the target edge's source and destination nodes.

3. *Node-Level* Tasks (e.g., node classification): Append the target node's token to the sequence.

The final token's output is fed to a randomly initialized multilayer perceptron (MLP). Fig. 2 (Appendix) illustrates the implementation.

The transformer weights are initialized from pre-trained checkpoints, and the MLP layers are initialized randomly. All parameters are updated during fine-tuning.

This formulation ensures seamless knowledge transfer from pre-training, mirroring successes in NLP (Brown et al., 2020; Wei et al., 2022).

# 3. Experiments

## 3.1. Datasets

Recent advances in AI for scientific discovery (Wang et al., 2023) motivate our evaluation of GraphGPT on large-scale scientific graph datasets spanning physics, chemistry, and bioinformatics. To demonstrate its versatility across graph tasks, we select benchmarks for graph-, edge-, and node-level objectives:

- Graph-level: PCQM4Mv2 (quantum chemistry), ogbg-molpcba (molecular property prediction) and Triangles (triangles counting).

- Edge-level: ogbl-ppa (protein-protein associations) and ogbl-citation2 (citation networks).

- Node-level: ogbn-proteins (protein interaction networks) and ogbn-arxiv (paper categorization).

Dataset statistics are detailed in Table 9 (Appendix A).

- PCQM4Mv2 contains $> 3.7$ million organic molecules from PubChemQC (Nakata & Shimazaki, 2017). Nodes represent atoms (9D attributes: atomic number, chirality, etc.), and edges denote chemical bonds (3D attributes: bond type, stereochemistry, conjugation).

- ogbg-molpcba is a smaller molecular dataset (Wu et al., 2017) with the same node/edge attributes.

- Triangles (Knyazev et al., 2019) contains 45k graphs (no node/edge attributes).

- ogbl-ppa: Nodes are proteins from 58 species; edges represent functional associations (Szklarczyk et al., 2019).

- ogbl-citation2: A directed citation network with $\sim 3$ million papers (nodes) and $> 30$ million edges.

- ogbn-proteins: Undirected, weighted graph of 132,534 proteins (nodes) with 8D edge attributes encoding association strengths.

- ogbn-arxiv: Citation network of 169,343 papers; tasks involve predicting 40 subject categories.

Empirical results demonstrate that SMTP pre-training achieves superior or comparable performance across all benchmarks. Unless otherwise specified, reported GraphGPT results utilize SMTP pre-training.

## 3.2. Graph-Level Tasks

*Table 1.* Results of the graph regression task on the PCQM4Mv2 dataset. The metric is mean absolute error (MAE), the smaller the better. 86% of the valid dataset is added to training after hyper-parameters selection. Superscript numbers indicate source references, while subscript numbers correspond to model variants in Table 11 (Appendix). The best results are in bold, and second-best are underlined. This notation convention applies to all subsequent tables.

| | Models | MAE ↓ Valid | Test | Params |
|---|---|---|---|---|
| GNN | GCN[1] | 0.1379 | 0.1398 | 2.0M |
| | GIN[2] | 0.1195 | 0.1218 | 3.8M |
| | GCN[1]-VN[3] | 0.1153 | 0.1152 | 4.9M |
| | GIN[2]-VN[3] | 0.1083 | 0.1084 | 6.7M |
| GT | TokenGT[4] | 0.0910 | 0.0919 | 48.5M |
| | Graphformer[5] | 0.0864 | N/A | 48.3M |
| | GPS-Deep[6] | 0.0852 | 0.0862 | 138.1M |
| | GPS++ (no 3D)[7] | 0.0818 | N/A | 40.0M |
| | GPTrans-L[8] | 0.0809 | 0.0821 | 86.0M |
| Ours | GraphGPT-M | 0.0827 | N/A | 37.7M |
| | GraphGPT-B$_{12}$ | 0.0807 | N/A | 113.6M |
| | GraphGPT-B$_{24}$ | **0.0793** | N/A | 227.3M |
| | GraphGPT-B$_{48}$ | **0.0792** | **0.0804** | 453.4M |

[1]Kipf & Welling (2017), [2]Xu et al. (2019), [3]Gilmer et al. (2017), [4]Kim et al. (2022), [5]Ying et al. (2021), [6]Rampásek et al. (2022), [7]Masters et al. (2023), [8]Chen et al. (2023b)

We evaluate GraphGPT on two molecular datasets where tasks involve predicting quantum chemical properties solely from 2D molecular graphs—a practical alternative to relying on 3D equilibrium structures. Specifically, PCQM4Mv2 predicts the HOMO-LUMO energy gap, and ogbg-molpcba predicts 128 binary molecular properties.

On PCQM4Mv2, GraphGPT achieves a test MAE of **0.0804**, significantly outperforming the previous SOTA (0.0821, Chen et al. (2023b)).

Compared to GTs like TokenGT, Graphformer, GPS, and GPTrans—which require handcrafted features or intricate architectures to encode structural information—GraphGPT attains superior performance without manual feature engineering. It also surpasses GNNs by a substantial margin (Table 1).

**Analysis** (Tables 1 and 2):

*Lossless Serialization*: The Eulerian path-based serializa-

tion and generative pre-training enable GraphGPT to fully capture structural and semantic graph information.

*Scalability*: While GTs with fewer parameters often plateau when scaled (Shi et al., 2022), GraphGPT shows consistent improvement up to **200M** parameters. The log-log scaling law plot for both pre-training loss and supervised fine-tuning loss is shown in Fig. 3 (Appendix).

*Parameter Efficiency*: GraphGPT's larger parameter count may reflect its capacity to implicitly learn features that other GTs encode manually. Generative pre-training also allocates model capacity to generation tasks, potentially limiting discriminative performance of models at smaller scales.

*Limitations*: Pre-training on additional external large-scale molecular datasets yielded diminishing returns, suggesting saturation in 2D structural information. Incorporating 3D molecular data could help address this limitation.

*Transfer Learning*: When fine-tuned on ogbg-molpcba, our PCQM4Mv2-pretrained model achieves results exceeding powerful GNNs (GCN, GIN) and matching SOTA GTs (Table 2).

*Table 2.* Results of the graph classification task on the ogbg-molpcba dataset. All the baseline results are from the OGB leaderboard or the corresponding papers. [†] indicates the model is pre-trained on PCQM4M-v2 dataset.

| Models | Average Precision (%) ↑ | | Params |
| | Test | Valid | |
| --- | --- | --- | --- |
| GCN[1] | $20.20_{\pm 0.24}$ | $20.59_{\pm 0.33}$ | 0.57M |
| GIN[2] | $22.66_{\pm 0.28}$ | $23.05_{\pm 0.27}$ | 1.92M |
| GINE[3]-VN[4] | $29.17_{\pm 0.15}$ | $30.65_{\pm 0.30}$ | 6.1M |
| NGIN[5]-VN[4] | $30.07_{\pm 0.37}$ | $30.59_{\pm 0.56}$ | 44.19M |
| PDF[6] | $30.31_{\pm 0.26}$ | $31.15_{\pm 0.20}$ | 3.84M |
| Graphormer-L[†7] | $31.40_{\pm 0.32}$ | $\underline{32.27_{\pm 0.24}}$ | 119.5M |
| EGT-Larger[†8] | $29.61_{\pm 0.24}$ | N/A | 110.8M |
| GRPE-Large[†9] | $31.50_{\pm 0.10}$ | N/A | 118.3M |
| GPTrans-L[†10] | $\mathbf{32.43_{\pm 0.22}}$ | N/A | 86.0M |
| GraphGPT-M[†] | $30.13_{\pm 0.25}$ | $31.62_{\pm 0.24}$ | 37.7M |
| GraphGPT-B[†]$_{12}$ | $31.28_{\pm 0.23}$ | $\underline{32.27_{\pm 0.15}}$ | 113.6M |
| GraphGPT-B[†]$_{24}$ | $\underline{31.81_{\pm 0.1}}$ | $\mathbf{32.54_{\pm 0.2}}$ | 227.3M |

[1]Kipf & Welling (2017), [2]Xu et al. (2019), [3]Brossard et al. (2020), [4]Gilmer et al. (2017), [5]Zhang & Li (2021), [6]Yang et al. (2023), [7]Ying et al. (2021), [8]Hussain et al. (2022), [9]Park et al. (2022), [10]Chen et al. (2023b)

### 3.2.1. GRAPH STRUCTURE UNDERSTANDING

To evaluate GraphGPT's ability to learn structural patterns through generative pre-training, we use the Triangles dataset with the task of counting triangles. The dataset is split into: 1). *Training/Validation*: 30k and 5k small graphs ($\leq 25$ nodes); 2). *Testing*: 5k small graphs (Test-small) and 5k large graphs (25–100 nodes, Test-large).

This task is challenging even for in-distribution (ID) graphs and considerably harder for out-of-distribution (OOD) graphs.

**Pre-Training Setup**: We augment pre-training with diverse datasets, *i.e.*, Reddit-threads (Rozemberczki et al., 2020), Erdős-Rényi random graphs (Erdos et al., 1960), and Internal real-world graphs (See Table 9, Appendix A).

**Analysis** (Table 3):

*Pre-Training Efficacy*: GraphGPT achieves comparable accuracy to GTs on ID graphs and superior OOD generalization (lower variance). This demonstrates that generative pre-training effectively encodes structural knowledge transferable to downstream tasks.

*Impact of Graph Types*: Pre-training on real-world graphs (e.g., internal datasets) outperforms random Erdős-Rényi graphs, suggesting meaningful structural patterns in real-world data enhance model learning.

*Dataset Diversity*: Combining Triangles with diverse datasets (Reddit-threads, internal graphs) yields better performance than pre-training on Triangles alone. This highlights the importance of diverse pre-training data for learning generalizable structural patterns.

*Attributed Graphs*: Models pre-trained on attributed graphs (PCQM4Mv2, ogbl-ppa, ogbn-proteins) and fine-tuned on Triangles achieve significant improvements: 64.3%/86.1%/86.6% vs. 32.6% (baseline GET without pre-training). This confirms that structural knowledge is obtained even when pre-training includes node/edge attributes.

### 3.3. Edge-Level Tasks

We evaluate GraphGPT on link prediction using the ogbl-ppa and ogbl-citation2 datasets. Results are summarized in Table 4.

*Performance Superiority*: GraphGPT significantly outperforms all baseline methods, including GNNs, heuristic models, and latent-factor approaches, across both datasets. This underscores the effectiveness of generative pre-training and sequence-based modeling for edge-level tasks.

*Scalability*: GraphGPT scales seamlessly to *2 billion parameters*, achieving sustained performance gains with increasing model size. This motivates future exploration of even larger architectures and datasets.

*Transformer Efficacy*: To our knowledge, GraphGPT is the first transformer-based model to achieve SOTA results on ogbl-ppa and ogbl-citation2, demonstrating the viability of sequence-driven architectures for large-scale edge-level tasks.

*Table 3.* Results of the graph classification task on the Triangles dataset. Superscript numbers indicate source references, while letters denote specific pre-training datasets. We report averaged metrics from 10 independent runs to ensure statistical reliability. The baseline results are from Müller et al. (2024).

| Models | Accuracy (%) ↑ | | Params |
| --- | --- | --- | --- |
| | T-small | T-large | |
| GIN[1] | $71.53_{\pm 0.94}$ | $33.54_{\pm 0.30}$ | 0.15M |
| Transformer[2] | $12.08_{\pm 0.31}$ | $10.01_{\pm 0.04}$ | 0.2M |
| Transformer-LapPE[3] | $78.29_{\pm 0.25}$ | $10.64_{\pm 2.94}$ | 0.2M |
| Transformer-RWSE[3] | $\mathbf{99.40_{\pm 0.10}}$ | $\underline{54.76_{\pm 7.24}}$ | 0.2M |
| Graphormer[4] | $\mathbf{99.09_{\pm 0.31}}$ | $42.34_{\pm 6.48}$ | 0.2M |
| GET-B | $32.60_{\pm 1.86}$ | $13.99_{\pm 1.78}$ | 113.5M |
| GraphGPT-B[a] | $92.16_{\pm 0.28}$ | $26.51_{\pm 1.01}$ | 113.5M |
| GraphGPT-B[b] | $81.38_{\pm 0.27}$ | $37.68_{\pm 0.99}$ | 113.5M |
| GraphGPT-B[c] | $\mathbf{99.08_{\pm 0.14}}$ | $38.80_{\pm 3.60}$ | 113.5M |
| GraphGPT-B[d] | $90.93_{\pm 0.51}$ | $40.79_{\pm 1.40}$ | 113.5M |
| GraphGPT-B[e] | $64.28_{\pm 0.33}$ | $17.38_{\pm 0.61}$ | 113.5M |
| GraphGPT-B[f] | $86.14_{\pm 7.38}$ | $26.94_{\pm 4.80}$ | 113.5M |
| GraphGPT-B[g] | $86.57_{\pm 2.74}$ | $23.45_{\pm 1.44}$ | 113.5M |
| GraphGPT-B[a+b] | $84.83_{\pm 0.81}$ | $39.62_{\pm 1.84}$ | 113.5M |
| GraphGPT-B[a+c] | $98.68_{\pm 0.18}$ | $50.07_{\pm 3.28}$ | 113.5M |
| GraphGPT-B[b+c] | $\underline{98.26_{\pm 0.30}}$ | $52.33_{\pm 2.61}$ | 113.5M |
| GraphGPT-B[a+b+d] | $89.98_{\pm 0.54}$ | $33.45_{\pm 2.51}$ | 113.5M |
| GraphGPT-M[a+b+c] | $95.07_{\pm 0.67}$ | $51.72_{\pm 1.12}$ | 33.7M |
| GraphGPT-B[a+b+c] | $98.63_{\pm 0.18}$ | $\mathbf{58.96_{\pm 1.90}}$ | 113.5M |

[1]Xu et al. (2019), [2]Vaswani et al. (2017), [3]Rampásek et al. (2022), [4]Ying et al. (2021)
Pre-trained with: [a]Triangles (45K), [b]Reddit-threads (0.2M), [c]Internal dataset (3.1M), [d]Random graphs (3.1M), [e]PCQM4M-v2 (3.7M), [f]ogbl-ppa (1), [g]ogbn-proteins (1).

## 3.4. Node-Level Tasks

We evaluate GraphGPT on two node-level benchmarks: ogbn-proteins predicts 112 binary protein function labels, and ogbn-arxiv classifies arXiv papers into 40 subject categories. Results are summarized in Table 5.

*ogbn-proteins*: GraphGPT surpasses well-tuned GNN baselines (GCN, GraphSAGE, GAT) and significantly outperforms graph transformers (GTs). Remarkably, GraphGPT achieves competitive performance with input subgraphs of $\sim 40$ nodes, while SOTA GNNs like AGDN (Sun et al., 2025) require subgraphs with $> 22,000$ nodes.

*ogbn-arxiv*: GraphGPT delivers performance comparable to or approaching SOTA graph transformers and optimized GNNs.

The strong performance with minimal neighborhood sampling suggests that generative pre-training effectively encodes global structural and semantic graph information into node token embeddings and transformer parameters. This contrasts with traditional GNNs, which rely on extensive

*Table 4.* Results of the link prediction task on the ogbl-ppa and ogbl-citation2 datasets.

| Models | ogbl-ppa | ogbl-citation2 |
| --- | --- | --- |
| | HR@100 (%) ↑ | MRR (%) ↑ |
| Common Neighbor | $27.65_{\pm 0.00}$ | $51.47_{\pm 0.00}$ |
| Adamic Adar | $32.45_{\pm 0.00}$ | $51.89_{\pm 0.00}$ |
| Resource Allocation[1] | $49.33_{\pm 0.00}$ | $51.98_{\pm 0.00}$ |
| Node2Vec[2] | $22.26_{\pm 0.83}$ | $61.41_{\pm 0.11}$ |
| Matrix Factorization[3] | $32.29_{\pm 0.94}$ | $51.86_{\pm 4.43}$ |
| GCN[4] | $18.67_{\pm 1.32}$ | $84.74_{\pm 0.21}$ |
| GraphSAGE[5] | $16.55_{\pm 2.40}$ | $82.60_{\pm 0.36}$ |
| SEAL[6] | $48.80_{\pm 3.16}$ | $87.67_{\pm 0.32}$ |
| AGDN[7] | $41.23_{\pm 1.59}$ | $85.49_{\pm 0.29}$ |
| SIEG[8] | $63.22_{\pm 1.74}$ | $90.18_{\pm 0.15}$ |
| MPLP[9] | $65.24_{\pm 1.50}$ | $90.72_{\pm 0.12}$ |
| RefinedGAE[10] | $\underline{73.74_{\pm 0.92}}$ | $84.55_{\pm 0.15}$ |
| GraphGPT-M | $65.44_{\pm 0.43}$ | $\underline{92.82_{\pm 0.27}}$ |
| GraphGPT-B | $68.76_{\pm 0.67}$ | $\mathbf{93.05_{\pm 0.20}}$ |
| GraphGPT-XXL | $\mathbf{76.55_{\pm 0.67}}$ | N/A |

[1]Zhou et al. (2009), [2]Grover & Leskovec (2016), [3]Mnih & Salakhutdinov (2008), [4]Kipf & Welling (2017), [5]Hamilton et al. (2017), [6]Zhang et al. (2021), [7]Sun et al. (2025), [8]Shi et al. (2024), [9]Dong et al. (2024), [10]Ma et al. (2024)

local aggregation with feature propagation.

## 3.5. Ablation Study

We analyze the impact of three core components of GraphGPT: pre-training, node re-indexing and node identity encoding.

### 3.5.1. PRE-TRAINING

The self-supervised NTP or SMTP tasks are central to GraphGPT's success. As shown in Table 6, pre-training delivers performance improvements of 10–100% across graph-, edge-, and node-level tasks. These gains highlight its role in enabling the model to learn intrinsic graph structural patterns and capture semantic relationships inherent in node and edge attributes.

### 3.5.2. NODE RE-INDEXING

As illustrated in Fig. 1, we re-index the nodes based on their order in the (semi-)Eulerian path. To evaluate the effectiveness of this approach, we conduct experiments on the ogbg-molpcba dataset, with results summarized in Tab. 7.

While node re-indexing increases pre-training loss, it consistently improves performance on downstream tasks across various model sizes. This technique acts as a form of data augmentation, preventing the model from memorizing graph-specific artifacts—such as arbitrary node label-

*Table 5.* Results of the node classification task on the ogbn-proteins and ogbn-arxiv datasets.

| Models | ogbn-proteins ROC-AUC (%) ↑ | ogbn-arxiv Accuracy (%) ↑ |
|---|---|---|
| GCN[1,2] | $77.29_{\pm 0.46}$ | $73.53_{\pm 0.12}$ |
| GraphSAGE[1,3] | $82.21_{\pm 0.32}$ | $73.00_{\pm 0.28}$ |
| GAT[1,4] | $85.01_{\pm 0.46}$ | $73.30_{\pm 0.18}$ |
| DRGAT[5] | N/A | $\mathbf{74.16_{\pm 0.07}}$ |
| AGDN[6] | $\mathbf{88.65_{\pm 0.13}}$ | $73.41_{\pm 0.25}$ |
| DeeperGCN[7] | $85.80_{\pm 0.17}$ | $71.92_{\pm 0.16}$ |
| GraphGPS[1,8] | $77.15_{\pm 0.64}$ | $71.23_{\pm 0.59}$ |
| NAGphormer[1,9] | $72.17_{\pm 0.45}$ | $70.88_{\pm 0.24}$ |
| Exphormer[1,10] | $77.62_{\pm 0.33}$ | $72.32_{\pm 0.36}$ |
| GOAT[1,11] | $79.31_{\pm 0.42}$ | $72.76_{\pm 0.29}$ |
| NodeFormer[1,12] | $77.86_{\pm 0.84}$ | $67.78_{\pm 0.28}$ |
| SGFormer[1,13] | $79.92_{\pm 0.48}$ | $72.76_{\pm 0.33}$ |
| Polynormer[1,14] | $79.53_{\pm 0.67}$ | $73.40_{\pm 0.22}$ |
| GraphGPT-S | $83.56_{\pm 0.16}$ | $70.83_{\pm 0.33}$ |
| GraphGPT-M | $84.02_{\pm 0.21}$ | $71.20_{\pm 0.34}$ |
| GraphGPT-B | $85.33_{\pm 0.10}$ | $72.10_{\pm 0.30}$ |

[1]Luo et al. (2024), [2]Kipf & Welling (2017), [3]Hamilton et al. (2017), [4]Vaswani et al. (2017), [5]Zhang et al. (2023), [6]Sun et al. (2025), [7]Li et al. (2023), [8]Rampásek et al. (2022), [9]Chen et al. (2023a), [10]Shirzad et al. (2023), [11]Kong et al. (2023), [12]Wu et al. (2022), [13]Wu et al. (2024), [14]Deng et al. (2024)

ing—and thereby enhancing generalization.

Furthermore, re-indexing enables constrained decoding of node tokens during graph generation with GraphGPT, reducing the search space for valid outputs.

### 3.5.3. NODE IDENTITY ENCODING

Node identity encoding (see §2.2.3)—representing nodes' identity in large graphs as multiple tokens—is critical for edge- and node-level tasks. Using GraphGPT-mini (a lightweight variant to conserve computational resources), we demonstrate that this method significantly enhances performance (Table 8). Further implementation details are provided in Appendices A and F.

## 4. Limitations

We critically assess the limitations of GraphGPT to contextualize its applicability and inspire future improvements.

***Transferability.*** GraphGPT's reliance on dataset-specific pre-training limits its ability to generalize across domains with divergent semantics (e.g., social networks vs. molecular graphs). However, it demonstrates robust *cross-dataset structural understanding* (§3.2.1) and effective *intra-domain transferability*, as evidenced by molecular data experiments (§3.2).

*Table 6.* Ablation study of pre-training on the datasets of various types of tasks. Superscripts $D/E$ stand for transformer decoder/encoder. ∗ means both molpcba and PCQM4Mv2 datasets are used for SMTP pre-training, and † indicates that the model is further trained using PCQM4M-v2's regression task. For the PCQM4Mv2 dataset, the metric is MAE, the lower the better.

| DATASETS | PRE-TRAINING | TEST | VALID |
|---|---|---|---|
| PCQM4MV2 | $\mathbf{X}^D$ | N/A | 0.0978 |
| | $\mathbf{X}^E$ | N/A | 0.0856 |
| | NTP | N/A | 0.0875 |
| | SMTP | N/A | **0.0807** |
| OGBG-MOLPCBA | $\mathbf{X}^D$ | 12.80 | 13.31 |
| | $\mathbf{X}^E$ | 25.80 | 26.33 |
| | NTP | 23.85 | 27.77 |
| | SMTP | 27.56 | 28.74 |
| | SMTP* | 27.20 | 28.49 |
| | SMTP* + FT† | **28.07** | **29.01** |
| OGBL-PPA | $\mathbf{X}^D$ | 41.28 | 40.14 |
| | $\mathbf{X}^E$ | 42.13 | 41.57 |
| | NTP | 55.56 | 54.87 |
| | SMTP | **55.68** | **54.93** |
| OGBN-PROTEINS | $\mathbf{X}^D$ | 57.52 | 61.19 |
| | $\mathbf{X}^E$ | 53.20 | 56.39 |
| | NTP | 75.61 | 80.47 |
| | SMTP | **83.56** | **87.73** |

*Table 7.* Ablation study of node re-indexing on the ogbg-molpcba dataset with two model sizes. PT means pre-training.

| PARAMS | RE-INDEX | PT LOSS | TEST | VALID |
|---|---|---|---|---|
| 4.48M | ✗ | **0.0844** | 23.10 | 25.25 |
| | ✓ | 0.0874 | **23.85** | **27.77** |
| 114.12M | ✗ | **0.0689** | 22.70 | 26.21 |
| | ✓ | 0.0750 | **25.17** | **28.57** |

***Dataset size.*** Performance on some small- to medium-sized datasets (e.g., ogbn-arxiv) lags behind traditional GNNs. This can be mitigated by expanding datasets with semantically aligned data.

***Computational Cost.*** Pre-training on large-scale graphs (ogbn-proteins, ogbl-ppa) or extensive small graphs (PCQM4Mv2) with 50M+ parameters is resource-intensive. For example, pre-training GraphGPT-B (100M+ parameters) on PCQM4Mv2 with $1 \times 10^9$ tokens requires $\sim 63$ V100 GPU hours, and fine-tuning incurs $\sim 3$ V100 GPU hours per epoch in the distributed data parallel setting with 4 GPUs.

While GraphGPT is less practical for small datasets due to compute-performance trade-offs, it excels with large-scale data. Emerging techniques like quantization (Dettmers et al., 2022; Frantar et al., 2022), distributed training frameworks (Rasley et al., 2020; Shoeybi et al., 2019), and transformer

*Table 8.* Ablation study of node identity encoding (NIE) on the ogbl-ppa and ogbn-proteins datasets.

| DATASETS | PARAMS | NIE | TEST | VALID |
|---|---|---|---|---|
| OGBL-PPA | 14.75M | ✗ | 44.38 | 45.08 |
| | | ✓ | **55.56** | **54.87** |
| OGBN-PROTEINS | 10.76M | ✗ | 60.22 | 65.66 |
| | | ✓ | **75.61** | **80.47** |

optimizations (Dao, 2024) are poised to alleviate these costs.

***Future Directions.*** These limitations highlight opportunities for research in cross-domain transfer, data-efficient training, and scalable architectures.

## 5. Related Works

***Graph Neural Networks (GNNs)*** GNNs have dominated graph learning for decades, with numerous variants achieving strong performance across tasks (Wu et al., 2020). However, they face fundamental limitations such as over-smoothing and over-squashing (Rusch et al., 2023; Alon & Yahav, 2021), which hinder their scalability and ability to model long-range dependencies.

***Graph Transformers (GTs)*** Inspired by transformers' success in NLP and CV, recent work has adapted these architectures to graphs (Ying et al., 2021; Rampásek et al., 2022; Müller et al., 2024). While GTs achieve competitive results on large-scale graph-level tasks (Müller et al., 2024), they typically rely on handcrafted structural features or GNN-based modules to encode graph topology—either in input representations (Ying et al., 2021; Kim et al., 2022; Masters et al., 2023) or attention mechanisms (Ying et al., 2021; Chen et al., 2022; Luo et al., 2023).

***Pre-training and fine-tuning*** The self-supervised pre-training and supervised fine-tuning paradigm, popularized by transformers (Vaswani et al., 2017), revolutionized NLP (Devlin et al., 2019; Radford et al., 2018). Scaling this approach with web-scale data (Brown et al., 2020) and techniques like instruction tuning (Wei et al., 2022) or reinforcement learning from human feedback (Ouyang et al., 2022) further advanced the field. In CV, self-supervised methods like MAE He et al. (2022) and MaskGIT (Chang et al., 2022) demonstrated that masked prediction tasks (e.g., reconstructing masked image patches) enable transformers to achieve SOTA results.

## 6. Conclusions

We introduce GraphGPT, a novel model built on the GET backbone, which achieves SOTA or near-SOTA performance across graph-, edge-, and node-level tasks on large-

scale benchmarks. By unifying pretext and downstream tasks into a sequence-based paradigm, GraphGPT demonstrates strong transferability in capturing both structural graph patterns and domain-specific knowledge (e.g., molecular properties). Notably, scaling GraphGPT to billions of parameters yields consistent performance gains, highlighting its potential as a foundation for graph-centric foundation models.

Looking ahead, GraphGPT's architecture is inherently scalable—capable of expanding to hundreds of billions of parameters—and offers promising avenues for integration or alignment with large language models (LLMs), bridging graph reasoning and textual intelligence.

## Acknowledgements

We thank our colleagues Guoshuai Wang and Tiange Xu for their insightful discussions and feedback. We appreciate the constructive feedback from the anonymous reviewers, which significantly improved the clarity of this paper. We thank our families for their unwavering support and encouragement throughout this project.

## Impact Statement

This paper presents work whose goal is to advance the field of Machine Learning. There are many potential societal consequences of our work, none which we feel must be specifically highlighted here.

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

# A. Datasets

The detailed statistics of the datasets are in the Tab. 9.

*Table 9.* Statistics of graph-/edge-/node-level datasets. Here 'BC' stands for binary classification. $p$ in Random Graph datasets means the probability of creating the edge between a node pair.

| datasets | # of graphs | avg # of nodes | avg # of edges | task-type | metrics |
|---|---|---|---|---|---|
| PCQM4Mv2 | 3,746,619 | 14.14 | 14.56 | regression | MAE |
| ogbg-molpcba | 437,929 | 26.0 | 28.1 | multi-label BC | AP |
| reddit-threads | 203,088 | 23.9 | 24.9 | BC | ROC-AUC |
| Triangles | 45,000 | 20.9 | 32.7 | multi-class classification | ACC |
| Internal dataset | 3,100,000 | 24.8 | 54.7 | N/A | N/A |
| Random Graph$_{p=0.03}$ | 3,100,000 | 67.1 | 74.8 | N/A | N/A |
| ogbl-ppa | 1 | 576,289 | 30,326,273 | BC | HR@100 |
| ogbl-citation2 | 1 | 2,927,963 | 30,561,187 | BC | MRR |
| ogbn-proteins | 1 | 132,534 | 39,561,252 | multi-label BC | ROC-AUC |
| ogbn-arxiv | 1 | 169,343 | 1,166,243 | multi-class classification | ACC |

## A.1. Subgraph Sampling

The subgraph sampling configurations for different datasets of large graphs are shown in the Tab. 10.

*Table 10.* Details of subgraph sampling for ogbl-ppa and ogbn-proteins datasets. 'seq-len' means the average length of the Eulerian sequences. Edge-ego means sampling subgraph around the central edge, and Node-ego means sampling around the central node.

| dataset | sampling | depth ($d$) | # neighbors ($n$) | seq-len |
|---|---|---|---|---|
| ogbl-ppa | edge-ego | 1 | 14 | 90 |
| | | 1 | 30 | 280 |
| ogbl-citation2 | edge-ego | 1 | 14 | 60 |
| | | 1 | 20 | 90 |
| | | 1 | 30 | 130 |
| ogbn-proteins | node-ego | 9 | 1 | 20 |
| | | 20 | 1 | 50 |
| | | 40 | 1 | 120 |
| | | 60 | 1 | 200 |
| ogbn-arxiv | node-ego | 1 | 30 | 30 |
| | | 1 | 40 | 40 |

# B. Models

We list the model specifics in the Tab. 11. We experiment with eight different scales of models.

# C. Implementation Details

## C.1. Graphs to Sequences of Tokens

The implementation uses PyTorch as the primary framework. For graph preprocessing tasks such as subgraph sampling, we utilize torch-geometric (Fey & Lenssen, 2019). When required, we employ NetworkX (Hagberg et al., 2008) to Eulerize (sub)graphs and identify (semi-)Eulerian paths. A custom tokenizer converts these paths into token sequences, with dataset-specific vocabularies constructed for each case.

*Table 11.* Statistics of GraphGPT models of different sizes. The GraphGPT-Base is of the same scale as Bert-Base (Devlin et al., 2019).

| Model-size | Hidden-size | # of layers | # of heads | Params (excluding embed) |
|---|---|---|---|---|
| Mini | 256 | 4 | 4 | 4.2M |
| S (Small) | 512 | 4 | 8 | 16.8M |
| M (Medium) | 512 | 8 | 8 | 33.6M |
| B / $B_{12}$ (Base) | 768 | 12 | 12 | 113.2M |
| $B_{24}$ (Base24) | 768 | 24 | 12 | 226.5M |
| $B_{48}$ (Base48) | 768 | 48 | 12 | 453.0M |
| L (Large) | 1024 | 24 | 16 | 402.7M |
| XXL (XXLarge) | 1600 | 48 | 25 | 2.0B |

### C.2. Model Backbone

We employ a transformer architecture based on Llama (Touvron et al., 2023), implemented via the Hugging Face Transformers library (Wolf et al., 2020), as the backbone for NTP pre-training. For SMTP pre-training, we modify the architecture by replacing the causal attention mask with a bidirectional attention mask to create an encoder. We initialize all parameters randomly, and train models at various scales (see Table 11).

### C.3. Training

The models are pre-trained and fine-tuned on A800-80G GPU clusters[5] using DeepSpeed's Stage-2 strategy with mixed precision (FP16/FP32) or BF16 (Rasley et al., 2020). We employ the AdamW optimizer (Loshchilov & Hutter, 2019) with a learning rate scheduler. To maximize computational efficiency in pre-training stage, we pack multiple graph sequences into single entries, optimizing context window utilization (Raffel et al., 2020) for certain graph datasets. Dataset-specific configurations are detailed in their respective sections below.

The variance is inherently low for most large-scale datasets (e.g., PCQM4Mv2, ogbl-ppa), where reporting variance is standard practice only when significant. For these datasets, 3–5 runs consistently yield minimal variance (as shown in tables). For the Triangles dataset, variance is higher—particularly on out-of-distribution (OOD) test data. So we conducted 10 runs to ensure robustness.

The pre-training and fine-tuning paradigm is illustrated in Fig. 2.

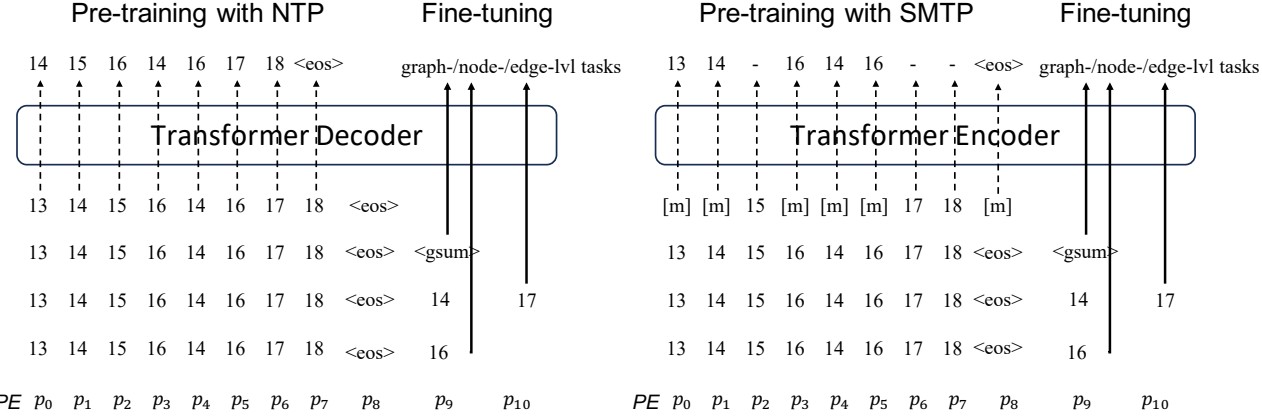

*Figure 2.* Pre-training and fine-tuning illustrations.

---

[5]We also utilize clusters of other types of GPUs, for example, Nvidia V100-32G, L20, L40 and etc.

## C.4. Vocabulary

In NLP, the vocabulary is typically constructed by tokenizing text data using the byte-pair encoding (BPE) algorithm (Sennrich et al., 2016). The resulting unique tokens form the vocabulary, which usually comprises frequent subwords from the text corpus.

In contrast, our GraphGPT employs a fundamentally different vocabulary construction approach. The vocabulary is split into two distinct parts: *1). structural and special tokens*, which are dataset-agnostic and transferable across different datasets. *2). semantic tokens*, which encode dataset-specific information, such as node and edge attributes.

An example is provided in Appendix F. In the graph sequence: *i).* tokens like '1', '2', etc., represent *structural tokens*; *ii).* Tokens such as 'ogbl-ppa#node#0#17' and 'ogbl-ppa#node#1#1959' are *semantics tokens*; *iii). Special tokens* like <gsum> and <eos> (Fig. 1) denote graph-specific functions (e.g., graph summary and end-of-sequence markers).

For the datasets ogbl-ppa/citation2, ogbn-proteins/arxiv, we set $k = 2$ (§2.2.3), resulting in vocabulary sizes of $41,634$ / $25,687$ and $31,360$ / $25,600$, respectively.

# D. Graphs to Sequences of Tokens

In this section, we show some examples of turning graphs to sequences of tokens.

## D.1. Molecular Graphs to Tokens

Below is one example of 2D molecular graphs in the ogbg-molpcba dataset in torch-geometric data format (Fey & Lenssen, 2019).

```
Data(x=[4, 9], edge_index=[2, 6], edge_attr=[6, 3], y=[128])
```

The graph has 4 nodes and 3 edges. The source and destination nodes of the edges are recorded in 'edge_index', and its dimension is $(2, 2 \cdot \text{number\_of\_edges})$ for undirected graphs. 'x' is the node attributes of 9 dimensions, and 'edge_attr' stores the edge attributes of 3 dimensions.

The node and edge attributes of the graphs are numbers. If we directly discretize them into tokens, i.e., using one token to represent each unique number, the numbers that appear few times in the dataset cannot be well-trained. At the same time, the vocabulary may blow up. Therefore, we split them into single digits and represent them with the combination of the following tokens. They are dataset agnostic, and can be shared across different datasets.

```
<->, <.>, <0>, <1>, <2>, <3>, <4>, <5>, <6>, <7>, <8>, <9>
```

The resulting vocabulary is 556 for both ogbg-molpcba and PCQM4Mv2.

Below shows the tokens from one of the possible (semi-)Eulerain paths of the above molecular graph.

```
['1', 'ogbg-molpcba#node#0#1', '<7>', 'ogbg-molpcba#node#2#1', '<1>', 'ogbg-
    molpcba#node#3#1', '<5>', 'ogbg-molpcba#node#6#1', '<1>', 'ogbg-molpcba#edge
    #0#1', '<1>', '2', '3', 'ogbg-molpcba#node#0#1', '<5>', 'ogbg-molpcba#node
    #2#1', '<4>', 'ogbg-molpcba#node#3#1', '<5>', 'ogbg-molpcba#node#4#1', '<3>',
     'ogbg-molpcba#node#6#1', '<2>', '2', 'ogbg-molpcba#node#0#1', '<5>', 'ogbg-
    molpcba#node#2#1', '<3>', 'ogbg-molpcba#node#3#1', '<5>', 'ogbg-molpcba#node
    #6#1', '<1>', '4', 'ogbg-molpcba#node#0#1', '<5>', 'ogbg-molpcba#node#2#1',
    '<4>', 'ogbg-molpcba#node#3#1', '<5>', 'ogbg-molpcba#node#4#1', '<3>', 'ogbg-
    molpcba#node#6#1', '<2>']
```

In the sequence of tokens above, for the node '1', we can deduce that its 9 dimensional attributes are $(7, 0, 1, 5, 0, 0, 1, 0, 0, 0)$. Node '1' is connected to '2' with edge attributes $(1, 0, 0)$. We set 0 as the the default value of the attributes in this dataset, and do not encode it into tokens.

In the (semi-)Eulerian path, a node may appear several times. We append its attributes tokens to one of its appearances randomly. This can prevent the model from copying the attributes from the previous appearance, and also shorten the resulting sequence.

For a graph obtained from Eulerization, an edge may present several times in the path. We apply the same logic to insert the edge attributes tokens.

As in the above sequence, node '2' appears two times, and its node attributes tokens are appended after its second appearance. There is no tokens encode the edge attributes of edge between '2' and '3', which implies the edge attributes are default value $(0, 0, 0)$.

### D.2. Subgraphs to Tokens

In edge/node-level tasks, we usually have one big graph. In this section, we use ogbl-ppa and ogbn-proteins datasets to show how to sample subgraphs from the big graph, and then transform the subgraph to sequences of tokens.

The whole ogbl-ppa dataset is summarized in torch-geometric format as follows.

```
Data(num_nodes=576289, edge_index=[2, 42463862], x=[576289, 58])
```

It has 576289 nodes and 21231931 edges in the training data. 'x' is the one-hot representation of the species that the node (protein) belongs to.

We sample a subgraph from it as below.

```
Data(num_nodes=30, root_n_id=[2], edge_index=[2, 84], x=[30, 2])
```

It has 30 nodes, 42 edges as in 'edge_index'. 'x' is the node attributes of 2 dimensions, and it encodes the node identity as described in Sec. 2.2.3. We partition the nodes (proteins) based on the associated species. The number of proteins inside each species varies from 616 to 41017. Finally we use 58 tokens for species and 41017 tokens for the local indices. Combined with the tokens for the structure and the special tokens, the total vocabulary is 41231.

Here 'root_n_id' records the two seed nodes, and the subgraph is sampled centered around them. The resulting tokens from one of the possible (semi-)Eulerian paths are:

```
['1', '2', '3', 'ogbl-ppa#node#0#17', 'ogbl-ppa#node#1#1959', '4', '5', 'ogbl-ppa
    #node#0#17', 'ogbl-ppa#node#1#2460', '6', '7', 'ogbl-ppa#node#0#17', 'ogbl-
    ppa#node#1#3566', '6', '8', 'ogbl-ppa#node#0#17', 'ogbl-ppa#node#1#4145',
    '6', '9', 'ogbl-ppa#node#0#20', 'ogbl-ppa#node#1#5334', '10', 'ogbl-ppa#node
    #0#27', 'ogbl-ppa#node#1#17324', '6', 'ogbl-ppa#node#0#17', 'ogbl-ppa#node
    #1#6850', '11', 'ogbl-ppa#node#0#17', 'ogbl-ppa#node#1#5498', '6', '12', '
    ogbl-ppa#node#0#17', 'ogbl-ppa#node#1#5776', '6', '4', 'ogbl-ppa#node#0#17',
    'ogbl-ppa#node#1#8183', '2', '5', '2', '13', 'ogbl-ppa#node#0#17', 'ogbl-ppa#
    node#1#3514', '2', 'ogbl-ppa#node#0#17', 'ogbl-ppa#node#1#9374', '14', 'ogbl-
    ppa#node#0#17', 'ogbl-ppa#node#1#6164', '15', 'ogbl-ppa#node#0#17', 'ogbl-ppa
    #node#1#8368', '2', '6', '16', 'ogbl-ppa#node#0#17', 'ogbl-ppa#node#1#10803',
     '6', '17', 'ogbl-ppa#node#0#17', 'ogbl-ppa#node#1#11465', '6', '10', '18', '
    ogbl-ppa#node#0#20', 'ogbl-ppa#node#1#16505', '6', '19', 'ogbl-ppa#node
    #0#17', 'ogbl-ppa#node#1#15071', '2', '20', 'ogbl-ppa#node#0#17', 'ogbl-ppa#
    node#1#7761', '2', '21', 'ogbl-ppa#node#0#17', 'ogbl-ppa#node#1#8828', '2',
    '22', 'ogbl-ppa#node#0#17', 'ogbl-ppa#node#1#14477', '2', '23', 'ogbl-ppa#
    node#0#17', 'ogbl-ppa#node#1#16026', '2', '24', 'ogbl-ppa#node#0#17', 'ogbl-
    ppa#node#1#16825', '6', '25', 'ogbl-ppa#node#0#17', 'ogbl-ppa#node#1#17615',
    '19', '25', '2', '26', 'ogbl-ppa#node#0#17', 'ogbl-ppa#node#1#19524', '2',
    '27', 'ogbl-ppa#node#0#17', 'ogbl-ppa#node#1#17854', '6', '28', 'ogbl-ppa#
    node#0#17', 'ogbl-ppa#node#1#17733', '6', '29', 'ogbl-ppa#node#0#27', 'ogbl-
    ppa#node#1#23255', '6', '30', 'ogbl-ppa#node#0#17', 'ogbl-ppa#node#1#19700',
    '6', '27', '1', 'ogbl-ppa#node#0#17', 'ogbl-ppa#node#1#20474']
```

In the ablation study on node identity encoding in Sec. 3.5.3, an example of the subgraph sampled from ogbl-ppa without identity encoding is shown below.

```
Data(num_nodes=30, root_n_id=[2], edge_index=[2, 136], x=[30, 1])
```

Different from the subgraph with node identity encoded in 'x', its node attribute 'x' contains only the information of the node's (protein) hosting species. It cannot be used to uniquely identify the nodes. The vocabulary decreases from 41231 to 214.

The resulting tokens from one of its possible (semi-)Eulerian paths is below.

```
['1', '2', '3', '4', 'ogbl-ppa#node#0#17', '5', '6', '7', '5', '8', '9', '1', '
    ogbl-ppa#node#0#17', '10', 'ogbl-ppa#node#0#17', '11', 'ogbl-ppa#node#0#17',
    '3', 'ogbl-ppa#node#0#17', '11', '12', '1', '5', '13', 'ogbl-ppa#node#0#17',
    '5', '14', 'ogbl-ppa#node#0#17', '5', '9', '10', '8', 'ogbl-ppa#node#0#17',
    '3', '15', 'ogbl-ppa#node#0#17', '3', '16', 'ogbl-ppa#node#0#17', '3', '2', '
    ogbl-ppa#node#0#20', '17', 'ogbl-ppa#node#0#27', '1', '18', 'ogbl-ppa#node
    #0#20', '1', '19', 'ogbl-ppa#node#0#17', '3', '9', 'ogbl-ppa#node#0#17',
    '20', 'ogbl-ppa#node#0#17', '10', '3', '21', '3', '5', '10', '12', 'ogbl-ppa#
    node#0#17', '3', '22', 'ogbl-ppa#node#0#17', '3', '17', '18', '3', '23',
    '13', '24', '5', '25', 'ogbl-ppa#node#0#17', '23', 'ogbl-ppa#node#0#17',
    '21', 'ogbl-ppa#node#0#17', '20', '5', '26', 'ogbl-ppa#node#0#17', '5', '22',
    '24', 'ogbl-ppa#node#0#17', '23', '5', '27', '6', 'ogbl-ppa#node#0#17',
    '28', 'ogbl-ppa#node#0#17', '7', 'ogbl-ppa#node#0#17', '28', '5', 'ogbl-ppa#
    node#0#17', '27', 'ogbl-ppa#node#0#17', '29', 'ogbl-ppa#node#0#17', '5',
    '30', 'ogbl-ppa#node#0#17', '5', '19', '5', '12', '20', '1']
```

In the following, we use the ogbn-proteins dataset as the example. The entire dataset is a large graph as below.

```
Data(num_nodes=132534, edge_index=[2, 79122504], edge_attr=[79122504, 8],
    node_species=[132534, 1], y=[132534, 112])
```

It has 132,534 nodes and 39,561,252 edges. 'node_species' stores the species' numeric id that the node (proteins) belongs to.

One sampled subgraph in the torch-geometric data format is:

```
Data(num_nodes=10, root_n_id=0, edge_index=[2, 22], edge_attr=[22, 8], y=[10,
    112], x=[10, 2])
```

It has 10 nodes, 11 edges as in 'edge_index'. Edge attributes is stored in 'edge_attr' of dimension 8. 'x' is the node attributes of 2 dimensions, and it encodes the node identity as described in Sec. 2.2.3. Its first dimension (token) represents the species, and the second is local numbering of each protein inside its species. Similar to the ogbl-ppa dataset, the identity encoding of 132,534 nodes occupies 25,465 tokens in the vocabulary, and the total vocabulary is 25,620.

'y' records the labels for the supervised node-level task. 'root_n_id' represents the target node, and the subgraph is sampled centered around it.

The resulting tokens from one of the possible (semi-)Eulerian paths are as follows.

```
['1', 'ogbn-proteins#node#0#3702', 'ogbn-proteins#node#1#16267', 'ogbn-proteins#
    edge#7#1', '<1>', '<6>', '<4>', '2', 'ogbn-proteins#node#0#3702', 'ogbn-
    proteins#node#1#6896', 'ogbn-proteins#edge#4#1', '<3>', '<4>', '<0>', '3', '
    ogbn-proteins#node#0#3702', 'ogbn-proteins#node#1#4121', 'ogbn-proteins#edge
    #4#1', '<3>', '<9>', '<8>', '4', 'ogbn-proteins#node#0#3702', 'ogbn-proteins#
    node#1#3963', 'ogbn-proteins#edge#4#1', '<1>', '<5>', '<3>', '5', 'ogbn-
    proteins#node#0#3702', 'ogbn-proteins#node#1#8259', 'ogbn-proteins#edge#4#1',
     '<4>', '<8>', 'ogbn-proteins#edge#7#1', '<2>', '<1>', '<5>', '6', '7', 'ogbn
    -proteins#edge#7#1', '<4>', '<1>', '<8>', '8', 'ogbn-proteins#node#0#3702', '
    ogbn-proteins#node#1#1', '7', 'ogbn-proteins#node#0#3702', 'ogbn-proteins#
    node#1#89', 'ogbn-proteins#edge#7#1', '<3>', '<2>', '<1>', '6', 'ogbn-
    proteins#node#0#3702', 'ogbn-proteins#node#1#955', 'ogbn-proteins#edge#7#1',
    '<2>', '<7>', '<0>', '9', 'ogbn-proteins#node#0#3702', 'ogbn-proteins#node
```

```
#1#7055', 'ogbn-proteins#edge#4#1', '<1>', '<6>', '<5>', '10', 'ogbn-proteins
#node#0#3702', 'ogbn-proteins#node#1#10010', 'ogbn-proteins#edge#4#1', '<1>',
'<6>', '<9>', '4', '5', 'ogbn-proteins#edge#4#1', '<2>', '<0>', '<7>', '3']
```

The original edge attributes are 8-dimensional vector of 3 decimal numbers from 0.001 to 1. We split them into single digits and represent them with the combination of the digits tokens as in App. E.

To reduce the number of tokens in the resultant sequences further, we multiply the number with 1000 and then minus it by 1. So we do not need to encode '.' any more. At the same time, we treat the value 0.001 (0 after the above transformation) as the default value and do not encode it with tokens.

## E. Graph-Level Task

### E.1. PCQM4M-v2

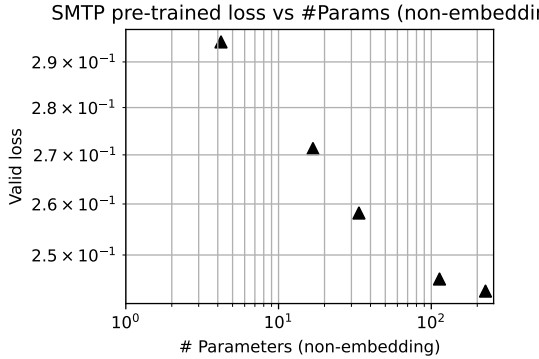 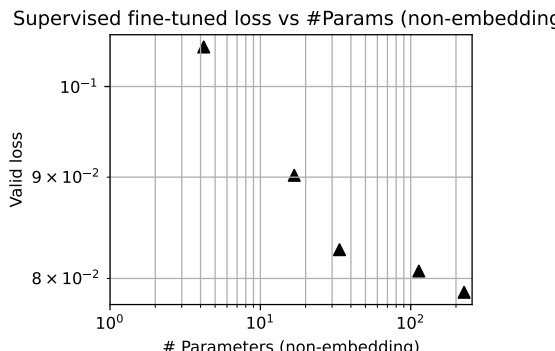

*Figure 3.* Log-log plot of pre-training loss and supervised fine-tuning loss versus the number of non-embedding parameters for the Mini/Small/Medium/Base/Base24 model configurations (see Table 11) on the PCQM4M-v2 dataset.

The pre-training and fine-tuning configurations for PCQM4M-v2 are in Tab. 12. The log-scale scaling law plot for both pre-training loss and supervised fine-tuning loss is shown in Fig. 3.

## F. Edge-Level Task

### F.1. ogbl-ppa

We use two tokens for the node identity encoding introduced in Sec. 2.2.3. Specifically, we use the species to partition the nodes, so the first token represents the species, and the second is the local indices of proteins inside each species.

The pre-training and fine-tuning configurations for ogbl-ppa are listed in Tab. 13. The loss of pre-training versus the number of tokens is shown in Fig. 4.

The fine-tuning data consists of subgraphs induced by the positive edges for training and equal negative edges randomly sampled.

In general, a larger model results in lower pre-training loss, and better results in down-stream fine-tuning tasks.

### F.2. ogbl-citation2

The pre-training and fine-tuning configurations for ogbl-citation2 are listed in Tab. 14. The subgraph sampling is edge-ego with $d = 1$ and $n = 14/30$ as in Tab. 10. The pre-training losses of the ogbl-citation2 dataset with different model sizes and subgraph sampling settings are shown in Fig. 5.

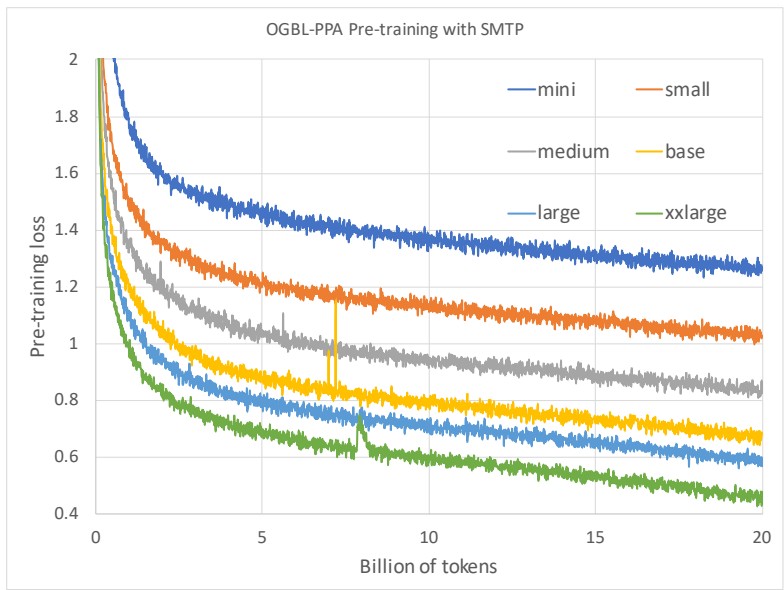

*Figure 4.* Pre-train loss versus tokens of ogbl-ppa dataset for models mini/small/medium/base/large/xxlarge as in Tab. 11.

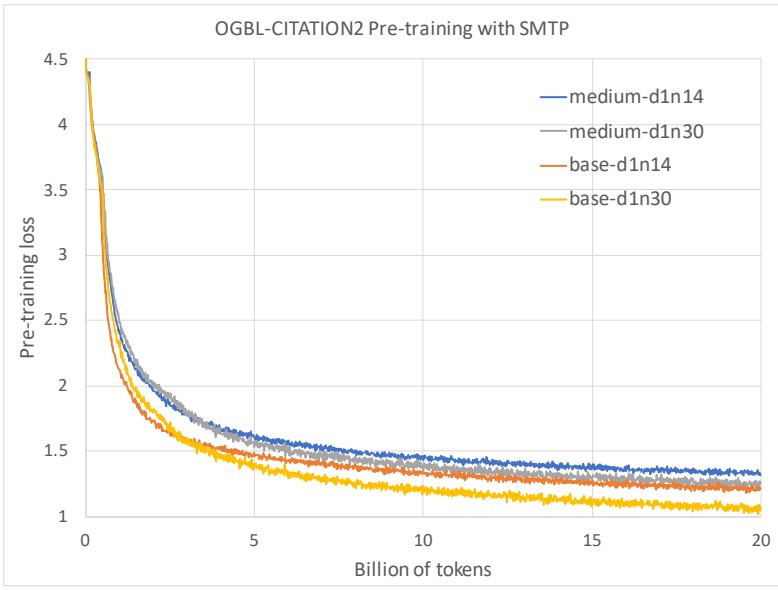

*Figure 5.* Pre-train loss versus tokens of ogbl-citation2 dataset for models medium/base as in Tab. 11.

*Table 12.* Pre-train and fine-tune configurations for the PCQM4M-v2 dataset. LSI means layer-scale-initialization, EMA is exponential moving average, MPE stands for max-position-embedding, and TWE means tie-word-embeddings.

| | pre-train | fine-tune |
|---|---|---|
| model-size | Mini Small Medium Base Base24 Base48 | |
| batch-size | 1024/1024/1024/1024/8192/8192 | 1024 |
| total | $1/1/1/1/4/4 \times 10^9$ tokens | 32 epochs |
| warmup | $10^8$ tokens | 9.6 epochs |
| lr scheduler | Warmup & linear decay | Warmup & cosine decay |
| max-lr | $3 \times 10^{-4}$ | $6/6/6/6/2/1.5 \times 10^{-4}$ |
| min-lr | 0 | automatic set |
| Adam-betas | $[0.9, 0.95]$ | $[0.9, 0.99]$ |
| Adam-eps | $1 \times 10^{-8}$ | $1 \times 10^{-10}$ |
| max-grad-norm | 5 | 1 |
| weight-decay | 0.1 | 0.02 |
| attention-dropout | 0.1 | |
| path-dropout | 0 | 0/0/0/0.05/0.1/0.2 |
| embed-dropout | 0 | |
| mlp-dropout | 0 | |
| LSI-val | NA | 1 |
| EMA | NA | |
| hidden-act | gelu | |
| MPE | 1024 | |
| TWE | FALSE | NA |

# G. Node-Level Task

## G.1. ogbn-proteins

The configurations of pre-training and fine-tuning are in Tab. 15. The subgraph sampling is node-ego with $d = 20$ and $n = 1$ as in Tab. 10. The node identity is encoded with two tokens similar to the ogbl-ppa in Sec. 3.3 (see App. F for details).

## G.2. ogbn-arxiv

The configurations of pre-training and fine-tuning are in Tab. 16. The subgraph sampling is node-ego with $d = 1$ and $n = 40$ as in Tab. 10. The node identity is encoded with two tokens.

# H. Question and Answering

**Q1. Evaluating GraphGPT on real-world citation networks (e.g., PubMed, Cora) or social networks (e.g., Twitter, Facebook graphs) could be great.**

*A1.* We evaluated GraphGPT on large-scale real-world citation networks: ogbn-arxiv (169K nodes, 1.17M edges) and ogbl-citation2 (2.93M nodes, 30.6M edges). These datasets are significantly larger than traditional benchmarks like Cora (2.7K nodes, 5.4K edges) and PubMed (19.7K nodes, 44.3K edges), aligning with our focus on scaling to massive graph data.

We chose these datasets because GraphGPT's performance benefits from large-scale pre-training data to learn inductive biases (e.g., node permutation invariance). For instance, pre-training on the small Triangles dataset (45K graphs) yielded poor fine-tuning results (32.6%), whereas scaling pre-training data improved performance to 99% (Section 3.2.1). This mirrors the trend in Vision Transformers (ViT), which outperform CNNs only with sufficiently large datasets (Dosovitskiy et al., 2021).

While GNNs may outperform GraphGPT on small datasets like Cora or PubMed, our goal is to demonstrate scalability for large-scale graphs—a critical challenge in modern applications.

**Q2. While GraphGPT enables a lossless and reversible graph-to-seq transformation, how well does it do this in**

*Table 13.* The Pre-training and fine-tuning configurations for the ogbl-ppa dataset. For XXL model, we use fp16 in the pre-train stage, and bf16 in the fine-tune stage for numerical stability.

| | pre-train | fine-tune |
|---|---|---|
| model-size | Mini Small Medium Base Large XXLarge | |
| batch-size | 1024 | 8192 |
| total | $2 \times 10^{10}$ tokens | 8/8/8/8/16/16 epochs |
| warmup | $10^9$ tokens | 2.4/2.4/2.4/2.4/4.8/4.8 epochs |
| lr scheduler | Warmup & linear decay | Warmup & cosine decay |
| max-lr | $3 \times 10^{-4}$ | $3/3/3/3/3/2 \times 10^{-5}$ |
| min-lr | 0 | automatic set |
| Adam-betas | $[0.9, 0.95]$ | $[0.9, 0.99]$ |
| Adam-eps | $1 \times 10^{-8}$ | $1 \times 10^{-10}$ |
| max-grad-norm | 5 | 1 |
| weight-decay | 0.1 | 0 |
| attention-dropout | 0.1 | |
| path-dropout | 0/0/0/0/0.1/0.1 | 0/0/0/0.05/0.1/0.2 |
| embed-dropout | 0 | |
| mlp-dropout | 0 | |
| LSI-val | NA | NA/NA/1/1/1/NA |
| EMA | NA | |
| hidden-act | gelu | |
| MPE | 1024 | |
| TWE | FALSE | NA |

**real-world noisy graphs?**

*A2.* While not the focus of this paper, we tested GraphGPT on an internal noisy graph dataset (3.1M graphs, avg. 24.8 nodes, 54.7 edges) for edge denoising.

Using a semi-supervised node classification task, GraphGPT achieved 10-20% F1 score improvement over baselines. We formulated the task analogously to Part-of-Speech tagging in NLP, leveraging token-level embeddings. The 'long' variant outperformed 'short' (see Fig. 1) likely due to its edge-agnostic token embeddings of nodes.

Results were robust enough for online deployment.

**Q3. What are the run-time comparisons with GNNs?**

*A3.* We evaluated the run-time of GraphGPT versus GNNs using the PCQM4M-v2 dataset on a single V100-32G GPU. The GNN baselines (adopted from Hu et al. (2021)) were implemented using the official GitHub repository[6].

Results are shown in Table 17. Run-time remains nearly constant across GraphGPT models ranging from 0.62M to 33.95M parameters. This consistency stems from an IO bottleneck during CPU-based data preprocessing. This time-consuming preprocessing phase involves determining if a graph is Eulerian, Eulerizing non-Eulerian graphs, and generating Eulerian paths.

Overall, GraphGPT's run-time is comparable to GNNs when model sizes are similar.

**Q4. What's the computational cost of GraphGPT models?**

*A4.* We have included computational cost details for the PCQM4Mv2 dataset in §4. The cost of other datasets are in the Tab. 18.

**Q5. How robust is the model to adversarial graph perturbations?**

*A5.* Adversarial robustness is a promising research area across NLP, CV, and graphs (Guo et al., 2021; Shao et al., 2022; Jin et al., 2020; Sun et al., 2023). While not our primary focus, preliminary results on noisy graphs (**Q2**) suggest robustness through large-scale training. A deeper study would bridge GraphGPT's transformer architecture with adversarial graph

---

[6]https://github.com/snap-stanford/ogb/tree/master/examples/lsc/pcqm4m-v2

*Table 14.* Pre-train and fine-tune configurations for the ogbl-citation2 dataset. We use bf16 in both the pre-training and fine-tuning stages for numerical stability. One epochs contains 10% randomly sampled positive edges and negative edges. For a given positive edge of head and tail node, we randomly sample a node as the tail node, and then form a negative edge with the head node.

| | pre-train | fine-tune |
|---|---|---|
| model-size | Medium Base | |
| batch-size | 1024 | 4096/2048 |
| total | $2 \times 10^{10}$ tokens | 32 epochs |
| warmup | $10^9$ tokens | 9.6 epochs |
| lr scheduler | Warmup & linear decay | Warmup & cosine decay |
| max-lr | $1 \times 10^{-4}$ | $3 \times 10^{-5}$ |
| min-lr | 0 | automatic set |
| Adam-betas | $[0.9, 0.95]$ | $[0.9, 0.99]$ |
| Adam-eps | $1 \times 10^{-8}$ | $1 \times 10^{-10}$ |
| max-grad-norm | 1 | |
| weight-decay | 0.1 | 0 |
| attention-dropout | 0.1 | |
| path-dropout | 0 | 0.05 |
| embed-dropout | 0 | |
| mlp-dropout | 0 | |
| LSI-val | N/A | |
| EMA | N/A | |
| hidden-act | gelu | |
| MPE | 512 | 1024 |
| TWE | FALSE | N/A |

defenses, an encouraging future direction.

**Q6. Can GraphGPT generate graphs that match real-world constraints (e.g., chemical validity)?**

*A6.* While generation is not the primary focus, preliminary experiments show GraphGPT can generate valid molecules after pre-trained on PCQM4M-v2.

However, generation quality depends on hyperparameters (e.g., temperature, top-p, iteration count T). Unconditional/conditional generation and diversity control require further study, which is planned for future work.

**Q7. Can you compare with other pre-trained-based graph models to highlight the advantages of GraphGPT?**

*A7.* While models like GraphBERT (Zhang et al., 2020), GraphMAE (Hou et al., 2022), and GCC (Qiu et al., 2020) employ graph pre-training, they primarily target small-scale datasets. GraphGPT's evaluation focuses on large-scale OGB leaderboard benchmarks, where existing pre-trained models lack competitive entries. Our comparisons align with state-of-the-art baselines dominating these leaderboards, emphasizing scalability and performance on real-world graph tasks.

**Q8. Why there is a need to introduce stochasticity when doing path identification?**

*A8.* GraphGPT lacks some inductive biases inherent to GNNs (e.g., node permutation invariance). Randomly sampling Eulerian paths per epoch forces the model to learn invariance between different paths of the same graph, akin to how ViT (lacking CNN's inductive biases) benefits from large-scale data and data augmentation. Empirically, this reduced overfitting on molecular datasets.

**Q9. Size to performance ratio.**

*A9.* We clarify parameter counts and performance across datasets below:

Graph-Level (Tab. 1–2): GraphGPT's parameter counts are comparable to prior SOTA (e.g., 113.6M vs. 86M for GPTrans).

Edge-Level (Tab.4): For ogbl-ppa, GraphGPT-B (145.3M) is a bit worse than Refined-GAE (295.8M), but GraphGPT-XXL (2B) achieves the highest performance. For ogbl-citation2, GraphGPT-M (46.8M) and GraphGPT-B (133.1M) outperform MPLP (749.8M).

*Table 15.* Configurations of pre-training with SMTP and fine-tuning for the ogbn-proteins dataset.

| | pre-train | fine-tune |
|---|---|---|
| model-size | \multicolumn{2}{c}{Small Medium Base} | |
| batch-size | 256 | 128 |
| total | $2 \times 10^{10}$ tokens | 16/16/8 epochs |
| warmup | $10^9$ tokens | 4.8/4.8/2.4 epochs |
| lr scheduler | Warmup & linear decay | Warmup & cosine decay |
| max-lr | $3 \times 10^{-4}$ | $3 \times 10^{-5}$ |
| min-lr | 0 | automatic set |
| Adam-betas | $[0.9, 0.95]$ | $[0.9, 0.99]$ |
| Adam-eps | $1 \times 10^{-8}$ | $1 \times 10^{-10}$ |
| max-grad-norm | \multicolumn{2}{c}{1} | |
| weight-decay | 0.1 | 0 |
| attention-dropout | \multicolumn{2}{c}{0.1} | |
| path-dropout | \multicolumn{2}{c}{0} | |
| embed-dropout | 0 | 0.1/0.2/0.1 |
| mlp-dropout | \multicolumn{2}{c}{0} | |
| LSI-val | \multicolumn{2}{c}{N/A} | |
| EMA | N/A | 0.999 |
| hidden-act | \multicolumn{2}{c}{gelu} | |
| MPE | \multicolumn{2}{c}{512} | |
| TWE | FALSE | N/A |

Node-Level (Tab.5): GraphGPT requires larger parameters on ogbn-proteins and ogbn-arxiv. This may reflect insufficient pre-training data for these tasks, leading to suboptimal parameter utilization.

**Q10. It is unclear what is intended by model scalability in §3.3; additionally scalability seems to not be the answer to the problem if we take into account costs and computational resources required to solve the tasks.**

*A10.* Our investigation of model scalability serves two critical purposes:

1. Studying performance limits reveals fundamental insights of data. Even small performance gains can reduce real-world validation costs (OGB Team, 2020).

2. This study aligns with foundational NLP scaling law research (Kaplan et al., 2020; Hoffmann et al., 2022), aiming to catalyze similar investigations for graph-structured data.

**Q11. How do you evaluate the correctness of the response? Do you query the Transformer model again with additional information if the response is not correct?**

*A11.* The model directly outputs predictions via the task head during inference. Results are evaluated using standard metrics (e.g., MAE, accuracy) for the downstream task. Each test/valid instance is processed once; no iterative querying is performed.

**Q12. How is the prompt structured? How do you express the task to solve?**

*A12.* We do not use prompts. Instead, tasks are encoded via specialized tokens appended to the input sequence and processed by an additional MLP head during fine-tuning as discussed in §2.3.2. Fig. 2 illustrates the implementations.

*Table 16.* Configurations of pre-training with SMTP and fine-tuning for the ogbn-arxiv dataset.

| | pre-train | fine-tune |
|---|---|---|
| model-size | Small Medium Base | |
| batch-size | 256 | 128 |
| total | $4 \times 10^9$ tokens | 4 epochs |
| warmup | $2 \times 10^8$ tokens | 1.2 epochs |
| lr scheduler | Warmup & linear decay | Warmup & cosine decay |
| max-lr | $3 \times 10^{-4}$ | $3/3/2 \times 10^{-4}$ |
| min-lr | 0 | automatic set |
| Adam-betas | $[0.9, 0.95]$ | $[0.9, 0.99]$ |
| Adam-eps | $1 \times 10^{-8}$ | $1 \times 10^{-10}$ |
| max-grad-norm | 1 | |
| weight-decay | 0.1 | 0 |
| attention-dropout | 0.1 | |
| path-dropout | 0 | 0/0/0.1 |
| embed-dropout | 0 | 0.1 |
| mlp-dropout | 0 | |
| LSI-val | N/A | |
| EMA | N/A | 0.9997 |
| hidden-act | gelu | |
| MPE | 1024 | |
| TWE | FALSE | N/A |

*Table 17.* Run-time comparison between GraphGPT variants and GNNs on the PCQM4Mv2 dataset. Time per epoch is measured in minutes.

| model | # params | time per epoch |
|---|---|---|
| GIN | 3.76 M | 9.25 min |
| GIN-virtual | 6.66 M | 11.2 min |
| GCN | 1.96 M | 8.0 min |
| GCN-virtual | 4.85 M | 9.6 min |
| GraphGPT-Tiny | 0.62 M | 20 min |
| GraphGPT-Mini | 4.39 M | 21 min |
| GraphGPT-Small | 17.17 M | 20 min |
| GraphGPT-Medium | 33.95 M | 20 min |
| GraphGPT-Base | 113.85 M | 46.7 min |

*Table 18.* Computational cost details of the main datasets in the paper. 'PT' means pre-training and 'FT' stands for fine-tuning. Time is measured in hours. The model size is 'Base' as in Tab. 11 with number of parameters about 110M. The corresponding hyper-parameters can be found in Tab. 13, 14, 15, 16.

| dataset | model size | PT time | FT time | GPU-PT | GPU-FT |
|---|---|---|---|---|---|
| ogbl-ppa | B | 58.73 h | 112.62 h | 8 Nvidia L20 | 16 V100-32G |
| ogbl-citation2 | B | 72 h | 100.3 h | 8 Nvidia L20 | 8 Nvidia L20 |
| ogbn-proteins | B | 27.1 h | 3.1 h | 8 Nvidia L20 | 1 V100-32G |
| ogbn-arxiv | B | 9.25 h | 4.3 h | 8 Nvidia L20 | 1 V100-32G |

