# OpenReview forum: "GraphGPT: Generative Pre-trained Graph Eulerian Transformer"
_ICML.cc/2025/Conference — ICML 2025 poster_

### Official Review · Reviewer_xS7R · 2025-03-12

**Overall Recommendation:** 4

**Summary:**

The paper introduces a self-supervised generative pre-trained model called GraphGPT based on a transformer architecture, which is called Graph Eulerian Transformer and employs a graph to sequence method which uses Eulerian paths. This method ensures reversibility in the graph-to-sequence translation. The transformer is first pre-trained for two kinds of self supervised tasks (next token prediction and scheduled masked token prediction) and then fine-tuned on downstream graph-, edge- and node-level tasks. The paper claims that the proposed method outperforms SOTA on Open Graph Benchmark datasets.

**Claims And Evidence:**

- “Randomly sample one valid path from possible candidates, introducing stochasticity as a data augmentation strategy akin to computer vision techniques (Perez & Wang, 2017).” -> It lacks explanation of why there is a need to introduce stochasticity when doing path identification

- “Introduce cyclic re-indexing:... where r is a random integer and N (hyperparameter) exceeds the maximum node count. This ensures uniform token training by distributing index frequencies.” -> It is not clear what is the relation between index frequencies and token training and not clear how re-indexing helps with that.

- “Connect components by adding synthetic edges between randomly selected nodes.” -> What is the criteria with which nodes are chosen for synthetic edges?

**Essential References Not Discussed:**

NA

**Experimental Designs Or Analyses:**

Please check the comments.

**Methods And Evaluation Criteria:**

It makes sense. However it would be better to expand and specify how the linear scheduling function for Mask Scheduling (SMTP) is implemented. Is it the same as the one proposed by Chang et. al? In case, specify.

**Other Comments Or Suggestions:**

Please see below.

**Other Strengths And Weaknesses:**

Strengths:

- Performance: demonstrates performance across multiple graph tasks, being close and sometimes higher than SOTA.
- Generalization: The experimental results are strong in many settings including datasets from different domains.

Weaknesses:

- Size to performance ratio: while it is able to compare to and sometimes beat SOTA, it needs to be noted that other models achieve the same performance with a number of parameters two orders of magnitude smaller than GraphGPT.

- In the benchmark only specific types of GraphGPT models are used. For instance in table 2, only GraphGPT B is used and not GraphGPT M.

- It is unclear what is intended by model scalability - section 3.3; additionally scalability seems to not be the answer to the problem if we take into account costs and computational resources required to solve the tasks.

**Questions For Authors:**

- In section 2.1 when talking about the implementation steps: why is there a need to do node reindexing?

- How do you choose which nodes to connect when adding synthetic edges?

- What linear scheduling function do you use for mask scheduling in SMTP?

- In section 3.3. What do you mean by “scales [...] to 2 billion parameters”? Can you elaborate on that?

- In section C.4, which k do you use for your token vocabulary (also with respect to section 2.2.3)

- How is the prompt structured? How do you express the task to solve?

- How do you evaluate the correctness of the response? Do you query the Transformer model again with additional information if the response is not correct?

**Relation To Broader Scientific Literature:**

I think this work contributes to some novel mechanisms to solve graph problems.

**Theoretical Claims:**

They seem to be correct.

---

> ### Author Rebuttal · Authors · 2025-04-01
>
> Thank you very much for the constructive feedback. Please let us know if there are any concerns that have not been addressed.
>
> Q1. Explanation of why there is a need to introduce stochasticity when doing path identification.
>
> A1: GraphGPT lacks some inductive biases inherent to GNNs (e.g., node permutation invariance).
>
> Randomly sampling Eulerian paths per epoch forces the model to learn invariance between different paths of the same graph, akin to how ViT (lacking CNN’s inductive biases) benefits from large-scale data and data augmentation.
>
> Empirically, this reduced overfitting on molecular datasets. While we did not include an explicit ablation study due to space constraints, we acknowledge its importance and will clarify this in the final version.
>
> Q2. Unclear relation between index frequencies and token training, and how re-indexing helps.
>
> A2: Without cyclic re-indexing, Eulerian paths would always start with low-index tokens (e.g., 0, 1, 2), leading to skewed token frequency distributions.
>
> Cyclic re-indexing randomizes starting indices (e.g., selecting from {0,1,…,255} for N=256), ensuring uniform training across all index tokens.
>
> This is critical for datasets like Triangles, where test graphs have significantly more nodes than training graphs (e.g., test graphs up to 100 nodes vs. training graphs ≤25 nodes). Without re-indexing, higher-index tokens (e.g., 25–255) remained undertrained, degrading performance.
>
> We will expand these details in the appendix.
>
> Q3. Criteria for choosing nodes for synthetic edges?
>
> A3: Synthetic edges are added to connect disconnected components. For example, if a graph has disconnected components A, B, and C, we connect A-B via a random node pair, then B-C similarly.
>
> The synthetic edges are tagged with a special token <edge_jump> to distinguish them from real edges. This ensures the graph becomes connected, enabling Eulerian path generation.
>
> We will clarify this in the text.
>
> Q4. Is SMTP linear scheduling function implementation the same as Chang et al.?
>
> A4: Yes, we use the linear scheduling function from MaskGIT (Chang et al., 2022), defined as γ(r) = 1 − r, where r ∈ [0, 1) and is uniformly distributed.
>
> We will explicitly state this in the final version.
>
> Q5. Size to performance ratio.
>
> A5: We clarify parameter counts and performance across datasets below:
>
> - Graph-Level (Tables 1–2): GraphGPT’s parameter counts are comparable to prior SOTA (e.g., 113.6M vs. 86M for GPTrans).
>
> - Edge-Level (Tab.4): For ogbl-ppa, GraphGPT-B (145.3M) is a bit worse than Refined-GAE (295.8M), but GraphGPT-XXL (2B) achieves the highest performance. For ogbl-citation2, GraphGPT-M (46.8M) and GraphGPT-B (133.1M) outperform MPLP (749.8M).
>
> - Node-Level (Tab.5): GraphGPT requires larger parameters on ogbn-proteins and ogbn-arxiv. This may reflect insufficient pre-training data for these tasks, leading to suboptimal parameter utilization.
>
> Q6. In the benchmark only specific types of GraphGPT models are used. For instance in table 2, only GraphGPT B is used and not GraphGPT M.
>
> A6: We tested models of size S/M/B for most datasets (e.g., PCQM4M-v2 and ogbl-ppa). Omitted results were excluded from the main text due to space constraints but did not alter the conclusions.
>
> These results will be added to the Appendix in the final version.
>
> Q7. model scalability-section 3.3.
>
> A7: Our investigation of model scalability serves two critical purposes:
>
> 1. Studying performance limits  reveals fundamental insights of data. Even small performance gains can reduce real-world validation costs [3].
>
> 2. This study aligns with foundational NLP scaling law research [1,2], aiming to catalyze similar investigations for graph-structured data.
>
> Q8. Why node reindexing in sec. 2.1?
>
> A8: Re-indexing nodes reduces overfitting. Ablation experiments confirm its effectiveness: re-indexing increased the training loss but improved validation/test performance.
>
> This ablation study, initially omitted for brevity, will be included in the Appendix.
>
> Q9. k value in sec. 2.2.3/C.4?
>
> A9: For the datasets ogbl-ppa/citation2, ogbn-proteins/arxiv, we set k=2, resulting in vocabulary sizes of 41,634, 25,687, 31,360, and 25,600, respectively.
>
> Q10. How is the prompt structured? How do you express the task to solve?
>
> A10: We do not use prompts. Instead, tasks are encoded via specialized tokens appended to the input sequence and processed by an additional MLP head during fine-tuning as discussed in sec. 2.3.2.
>
> A figure illustrating the implementations will be added to the Appendix in the final version.
>
> Q11. Correctness of the response? Iterative querying?
>
> A11: The model directly outputs predictions via the task head during inference. Results are evaluated using standard metrics (e.g., MAE, accuracy) for the downstream task. Each test/valid instance is processed once; no iterative querying is performed.
>
> [1] Kaplan et al., arxiv:2001.08361
>
> [2] Hoffmann et al., arxiv:2203.15556
>
> [3] https://ogb.stanford.edu/docs/linkprop/#ogbl-ppa

---

### Official Review · Reviewer_RhxR · 2025-03-12

**Overall Recommendation:** 3

**Summary:**

The paper presents GraphGPT, a novel self-supervised generative pre-trained model for graph learning that utilizes a new architecture called the Graph Eulerian Transformer (GET). The GET integrates a transformer architecture with a graph-to-sequence transformation method based on Eulerian paths, allowing for the reversible conversion of graphs into sequences of tokens representing nodes, edges, and attributes. The model is pre-trained using two self-supervised tasks: next-token prediction (NTP) and scheduled masked-token prediction (SMTP). GraphGPT is then fine-tuned for various downstream tasks, including graph, edge, and node-level predictions. The experimental results indicate that GraphGPT achieves state-of-the-art performance on multiple large-scale Open Graph Benchmark (OGB) datasets, particularly excelling in molecular property prediction and protein-protein interaction tasks. Notably, the model can scale to 2 billion parameters while maintaining performance increase, addressing scalability issues faced by traditional Graph Neural Networks (GNNs) and prior graph transformers. However, the paper could benefit from improved clarity in presentation, theoretical grounding, and methodological details to enhance its impact and applicability in diverse domains.

**Claims And Evidence:**

The authors provide extensive experimental results demonstrating that GraphGPT outperforms existing methods on various benchmark datasets. However, the claims regarding the theoretical underpinnings of the model needs to be further elaborated. For example, why the author choose the Eulerian path for graph-to-sequence transformation and how it ensures lossless and reversible mapping.

**Essential References Not Discussed:**

N/A

**Experimental Designs Or Analyses:**

The experimental designs and analyses appear sound, with a variety of datasets used to evaluate the model's performance across different tasks.

**Methods And Evaluation Criteria:**

The proposed methods and evaluation criteria are appropriate for the problem at hand. The use of self-supervised pre-training and the task-agnostic fine-tuning approach aligns well with the objectives of improving graph representation learning. The benchmarks selected for evaluation, including PCQM4Mv2 and ogbl-ppa, are relevant and widely recognized in the field.

**Other Comments Or Suggestions:**

- A preliminary section is recommended to provide a background and connection to the proposed method, which can help readers understand the motivation and significance of the work.
- The presentation of the methodology can be improved to better illustrate the proposed method.

**Other Strengths And Weaknesses:**

**Paper Strength**
1. Innovative methodology to convert graphs into sequences using Eulerian paths and using transformer architecture effectively capture graph structure.
2. Strong empirical findings demonstrating state-of-the-art performance across various benchmarks.
3. Comprehensive evaluation and analysis of the model's scalability and performance on large-scale datasets.

**Paper Weakness**
1. Presentation needs to be improved for clarity and readability. For example, how to conduct the re-index and cyclic re-index in the graph-to-sequence transformation?
2. The theoretical bounding of the model's claims is not sufficiently detailed. Why Eulerian path is chosen and how it ensures lossless and reversible mapping?
3. Methodology lacks clarity in certain aspects, particularly regarding tokenization and feature usage. How to handle non-text-based features is unclear.
4. The significance of transferability across different domains is not convincingly established.
5. The model did not compare with other pre-trained-based graph models to highlight the advantages of GraphGPT.

**Questions For Authors:**

1. Why did you choose the Eulerian path for graph-to-sequence transformation? How does it ensure lossless and reversible mapping?

2. How do you conduct the re-index and cyclic re-index in the graph-to-sequence transformation? What are their differences?

3. Can you provide more details on the tokenization process used, particularly for non-text-based features?

4. Can you compare with other pre-trained-based graph models to highlight the advantages of GraphGPT?

**Relation To Broader Scientific Literature:**

The contributions of this paper are well-positioned within the broader scientific literature on graph learning. The authors reference key prior works, situating their approach as an advancement in the adaptation of transformer architectures to graph data.

**Theoretical Claims:**

The paper does not present formal proofs for its theoretical claims, particularly regarding the lossless and reversible mapping of graphs to sequences using Eulerian paths.

---

> ### Author Rebuttal · Authors · 2025-04-01
>
> Thank you for your questions. Our responses are as follows:
>
> Q1. Why did you choose the Eulerian path for graph-to-sequence transformation? How does it ensure lossless and reversible mapping?
>
> A1: The Eulerian path was selected for its ability to traverse each edge exactly once, enabling a sequential representation that preserves graph topology without redundancy.
>
> As detailed in Section 2.2.1, this approach guarantees lossless and reversible mapping by construction, ensuring the sequence-to-graph conversion retains full structural fidelity. Theoretical justification is provided in the final paragraph of Section 2.2.1.
>
>
> Q2. How do you conduct the re-index and cyclic re-index in the graph-to-sequence transformation? What are their differences?
>
> A2: Implementation details for both methods are outlined in Section 2.2.1 (Node Re-indexing) and visualized in Figure 1. Re-indexing assigns fixed node IDs based on traversal order, while cyclic re-indexing dynamically rotates starting nodes to ensure uniform appearance frequencies of node-index tokens during training.
>
> Their comparative impacts on model performance are analyzed in A2 and A8 in our response to the 4th review.
>
>
> Q3. Can you provide more details on the tokenization process used, particularly for non-text-based features?
>
> A3: Non-text attributes (e.g., numerical or categorical features) are discretized into tokens via binning or directly, as described in Section 2.2.1 (Attribute Handling). Appendix D provides concrete examples. Notably, the benchmark datasets (OGB) lack text features, so our focus centers on structured numerical/categorical attribute processing.
>
>
> Q4. Can you compare with other pre-trained-based graph models to highlight the advantages of GraphGPT?
>
> A4: While models like GraphBERT [1], GraphMAE [2], and GCC [3] employ graph pre-training, they primarily target small-scale datasets. GraphGPT’s evaluation focuses on large-scale OGB leaderboard benchmarks, where existing pre-trained models lack competitive entries. Our comparisons align with state-of-the-art baselines dominating these leaderboards, emphasizing scalability and performance on real-world graph tasks.
>
>
> [1] Zhang et.al, GRAPH-BERT. arxiv:2001.05140
>
> [2] Hou et.al, GraphMAE. (KDD2022)
>
> [3] Qiu et.al, GCC. (KDD2020)

---

### Official Review · Reviewer_ndAQ · 2025-03-13

**Overall Recommendation:** 4

**Summary:**

The authors in this paper introduce GraphGPT for graph learning that leverages Graph Eulerian Transformer (GET). The proposed model uses a graph-to-sequence transformation method based on Eulerian paths, enabling it to convert graphs into token sequences for transformer-based processing.

**Claims And Evidence:**

C1: GraphGPT excels with large-scale data but there is a lack of how GNNs perform to compare with.

C2: While GraphGPT enables a lossless and reversible graph-to-seq transformation, how well does it do this in real-world noisy graphs?

**Essential References Not Discussed:**

None.

**Experimental Designs Or Analyses:**

Yes, the authors have very clearly demonstrated the performance of GraphGPT against several large-scale datasets that align with real-world applications.

**Methods And Evaluation Criteria:**

Yes, but it would be helpful to the readers if the authors could also include a runtime comparison with GNNs.

**Other Comments Or Suggestions:**

There are some minor typos in the paper, for example, in Section 1, "(semi- \n )Eulerian paths" that can be revised.

**Other Strengths And Weaknesses:**

This is a novel approach with scalability to billions of params. However, there is lack of clarity on the computational costs given the scalability and the robustness of the proposed model/method.

**Questions For Authors:**

Q1: How robust is the model to adversarial graph perturbations?

Q2: Can GraphGPT generate graphs that match real-world constraints (e.g., chemical validity)?

**Relation To Broader Scientific Literature:**

GraphGPT has been demonstrated to overcome the typical GNN limitations of over-smoothing and over-squashing and reduce the need for computing adjacency matrices by using Eulerian paths. It brings graph pre-training closer to transformers but may require more rigorous evaluation in interpretability, robustness, and graph generation quality.

**Theoretical Claims:**

Yes, but there is very little validation to test if the lossless property will also hold true in large, noisy, real-world datasets.

---

> ### Author Rebuttal · Authors · 2025-04-01
>
> Thank you very much for the valuable feedback. Our responses are as follows:
>
> Q1. GraphGPT excels with large-scale data but lacks comparisons with GNNs.
>
> A1: We compared GraphGPT with multiple GNN baselines (e.g., GCN, GIN, GCN-VN, GIN-VN) in all experiments. These baselines are standard in graph learning literature (e.g., [1, 2, 3]).
>
> Results are mostly organized in tables with GNN baselines, followed by graph transformer baselines, then GraphGPT.
>
> We will clarify this in table captions in the final version.
>
> Q2. While GraphGPT enables a lossless and reversible graph-to-seq transformation, how well does it do this in real-world noisy graphs?
>
> A2: While not the focus of this paper, we tested GraphGPT on an internal noisy graph dataset (3.1M graphs, avg. 24.8 nodes, 54.7 edges) for edge denoising.
>
> Using a semi-supervised node classification task, GraphGPT achieved 10-20% F1 score improvement over baselines. We formulated the task analogously to POS tagging in NLP, leveraging token-level embeddings. The "long" variant outperformed "short" (see Fig. 1) likely due to its edge-agnostic token embeddings of nodes.
>
> Results were robust enough for online deployment.
>
> Q3. Include runtime comparisons with GNNs.
>
> A3: Runtime comparisons are typically parameter-count based in literature. However, we will add runtime benchmarks for GNN baselines (from cited papers) and GraphGPT in the appendix.
>
> Q4. Yes, but there is very little validation to test if the lossless property will also hold true in large, noisy, real-world datasets.
>
> A4: The lossless property is theoretically guaranteed by Eulerian path theory, independent of noise.
>
> Empirical performance on noisy graphs (as in A2) demonstrates practical robustness.
>
> Q5. This is a novel approach with scalability to billions of params. However, there is lack of clarity on the computational costs given the scalability and the robustness of the proposed model/method.
>
> A5: Computational costs for PCQM4M-v2 are discussed in Section 4. We will expand appendix details for other datasets.
>
> Robustness: most results show low variance, indicating robustness across runs. For adversarial/noisy robustness, see A2 and A7.
>
> Q6. There are some minor typos in the paper, for example, in Section 1, "(semi- \n )Eulerian paths" that can be revised.
>
> A6: Corrected and will review other typo errors.
>
> Q7. How robust is the model to adversarial graph perturbations?
>
> A7. Adversarial robustness is a promising research area across NLP, CV, and graphs [4-7]. While not our primary focus, preliminary results on noisy graphs (A2) suggest robustness through large-scale training. A deeper study would bridge GraphGPT’s transformer architecture with adversarial graph defenses, an encouraging future direction.
>
> Q8. Can GraphGPT generate graphs that match real-world constraints (e.g., chemical validity)?
>
> A8. While generation is not the primary focus, preliminary experiments show GraphGPT can generate valid molecules after pre-trained on PCQM4M-v2.
>
> However, generation quality depends on hyperparameters (e.g., temperature, top-p, iteration count T). Unconditional/conditional generation and diversity control require further study, which is planned for future work.
>
>
> References
>
> [1] Chen et al. GPTrans (IJCAI 2023),
>
> [2] Masters et al. GPS++ (TMLR 2023),
>
> [3] Hussain et al. Triplet Interaction (ICML 2024),
>
> [4] Guo et al. Gradient-based Adversarial Attacks (EMNLP 2021),
>
> [5] Shao et al. On the Adversarial Robustness of ViT (NeurIPS 2022 workshop),
>
> [6] Jin et al. Adversarial Attacks and Defenses on Graphs (SIGKDD Explorations),
>
> [7] Sun et al. Adversarial Attack and Defense on Graph Data (IEEE TKDE2023)

---

> > ### Comment · Reviewer_ndAQ · 2025-04-02
> >
> > Thank you for addressing my questions and concerns. I will maintain my recommendation.

---

### Official Review · Reviewer_ozCk · 2025-03-14

**Overall Recommendation:** 3

**Summary:**

The paper "GraphGPT: Generative Pre-trained Graph Eulerian Transformer" proposes GraphGPT, a self-supervised generative pre-trained model for graph learning. The core contribution is the Graph Eulerian Transformer (GET), which enables transformers to process graph-structured data efficiently by converting graphs into sequence representations using Eulerian paths.

**Claims And Evidence:**

Authors mentioned that "GraphGPT scales to over 2 billion parameters with sustained performance gains."​, but some plots about the scaling laws would make this claim stronger.

**Essential References Not Discussed:**

No

**Experimental Designs Or Analyses:**

1. No mention of computational resources.
2. Three runs might be too few for high-variance tasks.

**Methods And Evaluation Criteria:**

Yes. But evaluate GraphGPT on real-world citation networks (e.g., PubMed, Cora) or social networks (e.g., Twitter, Facebook graphs) could be great.

**Other Comments Or Suggestions:**

N/A

**Other Strengths And Weaknesses:**

The structure of the paper is kind of messy, which may make audience harder to follow. Also the the model has too many footnotes in the table, I spend some time to figure out the meaning of each footnotes. It would be helpful to mention the meaning in the caption, or refer to the definition of footnotes.

**Questions For Authors:**

N/A

**Relation To Broader Scientific Literature:**

1. GNNs struggle with long-range dependencies due to repeated message passing, leading to over-smoothing and over-squashing. GraphGPT circumvents this limitation by tokenizing graphs into sequences via Eulerian paths, enabling transformer-based models to process entire graphs without localized message passing.

2. GraphGPT extends self-supervised pretraining techniques from NLP (e.g., BERT, GPT-3) to graphs. It introduces Next-Token Prediction (NTP) and Scheduled Masked-Token Prediction (SMTP), adapting masked language modeling (MLM) techniques for graphs.

**Theoretical Claims:**

Correct

---

> ### Author Rebuttal · Authors · 2025-04-01
>
> Thanks for the constructive feedback. Our replies are as follows:
>
> Q1. Authors mentioned that "GraphGPT scales to over 2 billion parameters with sustained performance gains.", but some plots about the scaling laws would make this claim stronger.
>
> A1: We appreciate the suggestion to strengthen our scaling analysis.
>
> Unlike NLP data, graph datasets lack uniformity, making it impractical to pre-train a single model across diverse domains (e.g., social networks vs. molecular graphs). As a result, GraphGPT is pre-trained and fine-tuned separately for different domains.
>
> For datasets like PCQM4Mv2, we observe performance saturation at 227M parameters (Table 1), while for ogbl-ppa, we scale up to 2B parameters. The pre-training loss decreases steadily with increasing model size (Figure 3, Appendix, page 19), mirroring trends in NLP scaling studies (e.g., Fig.1 of Llama1 [1]). Fine-tuning results for three model sizes are reported in Table 4 for brevity, but we will include results for all six sizes (4M–2B parameters) in the Appendix.
>
> While a comprehensive scaling law analysis (e.g., estimating model/data scaling exponents) is beyond this paper’s scope, we will add logarithmic plots of pre-training loss and fine-tuning metrics versus non-embedding parameter counts to the Appendix, analogous to NLP scaling plots (e.g., Figure 1 of [2]).
>
> Q2. Yes. But evaluate GraphGPT on real-world citation networks (e.g., PubMed, Cora) or social networks (e.g., Twitter, Facebook graphs) could be great.
>
> A2: We evaluated GraphGPT on large-scale real-world citation networks: ogbn-arxiv (169K nodes, 1.17M edges) and ogbl-citation2 (2.93M nodes, 30.6M edges). These datasets are significantly larger than traditional benchmarks like Cora (2.7K nodes, 5.4K edges) and PubMed (19.7K nodes, 44.3K edges), aligning with our focus on scaling to massive graph data.
>
> We chose these datasets because GraphGPT’s performance benefits from large-scale pre-training data to learn inductive biases (e.g., node permutation invariance). For instance, pre-training on the small Triangles dataset (45K graphs) yielded poor fine-tuning results (32.6%), whereas scaling pre-training data improved performance to 99% (Section 3.2.1). This mirrors the trend in ViT, which outperform CNNs only with sufficiently large datasets [3].
>
> While GNNs may outperform GraphGPT on small datasets like Cora or PubMed, our goal is to demonstrate scalability for large-scale graphs—a critical challenge in modern applications. We will clarify this rationale in the final version.
>
>
> Q3. No mention of computational resources.
>
> A3: We have included computational resource details for the PCQM4Mv2 dataset in Section 4.2 (Limitations: Computational Cost).
>
> To address this feedback, we will expand this discussion in the revised version to provide comprehensive resource metrics (e.g., GPU types, training time, memory usage) for all key experiments, including ogbn-proteins/arxiv and ogbl-ppa/citation2 datasets.
>
> This information will be added to the Appendix to ensure transparency.
>
> Q4. Three runs might be too few for high-variance tasks.
>
> A4: We clarify that variance is inherently low for most large-scale datasets (e.g., PCQM4Mv2, ogbl-ppa). For these datasets, 3–5 runs consistently yield minimal variance (as shown in tables). (It is common practice not to report the variance for PCQM4Mv2.)
>
> For the Triangles dataset, variance is higher—particularly on OOD test data. So we conducted 10 runs to ensure robustness. As shown in Table 3, GraphGPT pre-trained on large-scale data achieves superior performance with reduced variance (e.g., 58.96 ± 1.9 vs. 54.76 ± 7.24).
>
> To improve clarity, we will explicitly state the number of runs in table captions or footnotes where applicable.
>
> Q5. The structure of the paper is kind of messy.
>
> A5: We appreciate your input and welcome specific suggestions to improve clarity.
>
> Could you clarify which aspects of the structure are most problematic (e.g., section organization, flow of technical details in Sections 2.2–2.3, or appendices)?
>
> For instance, if the nested content in Sections 2.2 and 2.3 is unclear, we will reorganize subsections to enhance logical progression.
>
> We are committed to refining the structure to improve accessibility for readers.
>
> Q6. Too many footnotes in the table.
>
> A6: We apologize for the confusion caused by the table’s formatting.
>
> To conserve space and ensure clear citations, we used numerical superscripts to reference source papers (similar to [4]) and subscripts to denote model sizes (detailed in Appendix Tab. 10).
>
> We agree that this notation requires clarification and will incorporate explicit definitions of these notations directly into the table captions in the revised version.
>
> [1] Touvron et al., LLaMA 2023
>
> [2] Kaplan et al., Scaling laws 2020
>
> [3] Dosovitskiy et al., ViT. (ICLR2020)
>
> [4] Hussain et al., Triplet Interaction (ICML2024)

---

> > ### Comment · Reviewer_ozCk · 2025-04-04
> >
> > Thanks authors for the rebuttal, my concerns has been solved, after considering with other rebuttals, I will recommand this paper to be accepted.

---

### Decision · Program_Chairs · 2025-05-01

**Decision:**

Accept (poster)

**Comment:**

The paper proposes a self-supervised generative pre-trained model based on a transformer architecture that employs a graph to sequence method based on Eulerian paths. This helps overcoming the common problem of GNNs to capture long-range dependencies. All reviewers agree that the approach is innovative and that the results are convincing, and overall recommend to accept the paper.